# Satb2 neurons in the parabrachial nucleus mediate taste perception

Brooke C. Jarvie[1,2,4], Jane Y. Chen [1,2,4], Hunter O. King[1] & Richard D. Palmiter[1,2,3✉]

The neural circuitry mediating taste has been mapped out from the periphery to the cortex, but genetic identity of taste-responsive neurons has remained elusive. Here, we describe a population of neurons in the gustatory region of the parabrachial nucleus that express the transcription factor Satb2 and project to taste-associated regions, including the gustatory thalamus and insular cortex. Using calcium imaging in awake, freely licking mice, we show that Satb2 neurons respond to the five basic taste modalities. Optogenetic activation of these neurons enhances taste preferences, whereas chronic inactivation decreases the magnitude of taste preferences in both brief- and long-access taste tests. Simultaneous inactivation of Satb2 and calcitonin gene-related peptide neurons in the PBN abolishes responses to aversive tastes. These data suggest that taste information in the parabrachial nucleus is conveyed by multiple populations of neurons, including both Satb2 and calcitonin gene-related peptide neurons.

[1] Departments of Biochemistry and Genome Sciences, University of Washington, Seattle, WA 98195, USA. [2] Graduate Program in Neuroscience, University of Washington, Seattle, WA, USA. [3] Howard Hughes Medical Institute, University of Washington, Seattle, WA, USA. [4] These authors contributed equally: Brooke C. Jarvie, Jane Y. Chen. ✉email: palmiter@uw.edu

Taste allows animals to rapidly evaluate what they eat and drink. This helps guide them to seek and ingest foods that are nutritive and avoid potentially harmful or toxic foods. The five basic taste qualities include sweet, sour, salty, bitter, and umami and can be divided into multiple components, including taste identity (e.g., this is bitter), the value of the taste (bitter is aversive), and the intensity of the taste[1–3]. The value of a taste, such as salt during sodium deficiency, can change depending on concentration or the need state of an animal[4]. Furthermore, animals can learn to associate tastes with foods that have previously made them ill[5]. Electrophysiological recordings in different brain regions have been used to examine how these taste characteristics are encoded[3,6]. However, the genetic identity of taste-responsive cells throughout the brain has not been determined, which has limited the ability to examine their roles in taste-related behaviors using contemporary neuron-manipulation techniques.

There is evidence that some taste information is processed via labeled lines, where information about each taste quality follows a discrete pathway from taste receptors in the tongue to the cortex, with neuronal populations dedicated to that specific taste in each step of the neural circuit[3]. In the periphery, the majority of taste-responsive neurons respond to a single taste quality[7]. Furthermore, sweet and bitter tastes are relayed via separate pathways, although there are conflicting reports about whether there are spatially distinct regions of the cortex associated with each taste[8–10]. However, electrophysiological recordings from neurons throughout the central taste circuit show more complex responses. There are many neurons that preferentially respond to single taste qualities, but other neurons respond robustly to multiple tastes[3,6,11]. There are also neurons that conditionally change their response pattern depending on the state of the animal; for example, there are fewer salt-best and more sweet-best neurons during sodium depletion[11,12].

Taste is detected by taste receptors located on the tongue and epithelial palate, which relay information via cranial nerves to the rostral nucleus of the solitary tract (NTS) in the hindbrain[3,6]. In rodents, taste information is then sent to the parabrachial nucleus (PBN) in the pons, where it ascends to the insular cortex (IC) via the gustatory thalamus (parvicellular part of ventral posteromedial thalamus; VPMpc). As the secondary relay in the ascending taste pathway, the PBN acts as an interface between the hindbrain and forebrain for taste processing. The PBN is divided into nuclei based on cytoarchitecture and function[13,14]. The medial and lateral portions are separated by a white matter tract called the superior cerebellar peduncle (scp), while the waist region contains neurons intermingled within the scp[13]. Input from the gustatory region of the NTS innervates all three regions of the PBN, each of which contains neurons that respond to the five basic tastes[1,15]. The taste-responsive neurons appear to retain some topographic organization with more sweet-responsive neurons in the waist and medial PBN, while salt- and bitter-responsive neurons have been found primarily in the lateral PBN[14,15]. In addition to sending projections to the gustatory thalamus, the PBN also targets a series of limbic structures that are hypothesized to be important for taste-related tasks and reward[6]. These include the lateral hypothalamus (LH), bed nucleus of the stria terminalis (BNST), and central amygdala (CeA).

Despite the key placement of the PBN in the taste pathway, ablation of this region only partially disrupts taste preferences and taste reactivity without creating an ageusic animal[16,17]. These lesions increase acceptance of bitter tastes[18,19] and blunt responses to sweet and salty solutions under various conditions[20,21]. PBN lesions can also disrupt other taste-related behaviors such as need-driven sodium appetite and conditioned taste aversion (CTA)[22], although it is unclear how these lesions affect taste processing per se. Within the PBN, CGRP-expressing neurons within the external lateral region are critical for CTA[23,24], while another population of PBN neurons is involved in sodium appetite[25].

Here, we show that Satb2 neurons respond to the five basic taste modalities and innate taste preferences can be modulated bidirectionally by modulation of these neurons. Simultaneous inactivation of Satb2 and CGRP neurons in the PBN abolishes responses to aversive tastes. These data suggest that taste information in the PBN is conveyed by multiple populations of neurons, including both Satb2 and CGRP neurons. In addition to taste, the PBN receives nociceptive[26], thermosensory[27], visceral[14,28,29], and olfactory information[30], making it difficult to parse out the role of the PBN in mediating taste without affecting other signaling pathways. Tools that allow selective manipulation of taste-responsive neurons at each node in the circuit will facilitate further progress.

## Results

**The gustatory PBN contains Satb2-expressing neurons**. From the Allen Brain Atlas, we identified the transcription factor Satb2 as a potential marker for gustatory neurons in the PBN due to its expression primarily in the ventral–lateral, medial, and waist regions of the PBN; immunostaining for Satb2 confirmed the in situ hybridization data (Fig. 1a). To examine the role of these neurons in processing taste information, we generated a mouse line with Cre recombinase targeted to the *Satb2* locus by inserting an IRES-Cre:GFP cassette just beyond the termination codon (Supplementary Fig. 1a). We validated the line by injecting an adeno-associated virus carrying Cre-dependent YFP (AAV1-DIO-YFP) into the PBN of *Satb2^Cre* mice and determined that most YFP-expressing cells were immunoreactive for Satb2 (89.0 ± 1.6%, $n = 5$). Furthermore, the virus transduced 74 ± 3.1% of Satb2 neurons throughout the PBN ($n = 3$; Supplementary Fig. 1b, c).

We next determined that Satb2 neurons send projections to regions known to be involved in taste by labeling their terminal fields using AAV1-DIO-synaptophysin:mCherry (Fig. 1b and Supplementary Fig. 2a). Satb2 neurons project to ipsilateral and contralateral taste-related regions, although the labeling is much stronger on the ipsilateral side. These regions include the medial portion of the VPMpc, IC, medial region of CeA, LH, and BNST, as well as the arcuate nucleus, basomedial amygdala, and dorsomedial hypothalamus (Fig. 1b and Supplementary Fig. 2a). An important pathway for taste processing goes from the PBN to the VPMpc and then to the IC[3,6]. To determine whether Satb2 neurons are part of this circuit, we injected the retrograde tracer cholera-toxin B subunit conjugated to Alexa Fluor 647 (CTb-647) into the IC to label neurons in the VPMpc (Supplementary Fig. 2b). In combination, we targeted the excitatory Gq-coupled receptor hM3Dq to Satb2 neurons in the PBN and stimulated hM3Dq with an intraperitoneal injection of clozapine N-oxide (CNO, 1 mg/kg). Satb2–neuron activation resulted in Fos expression in ~35% of the fluorescent neurons in the VPMpc that project to the IC (Supplementary Fig. 2c, d). Furthermore, optogenetic stimulation of Satb2–neuron axons within the VPMpc (but not CeA or BNST) promotes licking behavior[31].

**Satb2 neurons respond to multiple tastes**. To determine whether Satb2 neurons in the PBN respond to taste, we unilaterally injected a virus that drives expression of a Cre-dependent fluorescent calcium indicator (AAV-DJ-DIO-GCaMP6s) along with AAV1-DIO-hM3Dq:mCherry into the PBN of *Satb2^Cre* mice[32]. We then implanted a gradient index (GRIN) lens over the PBN to

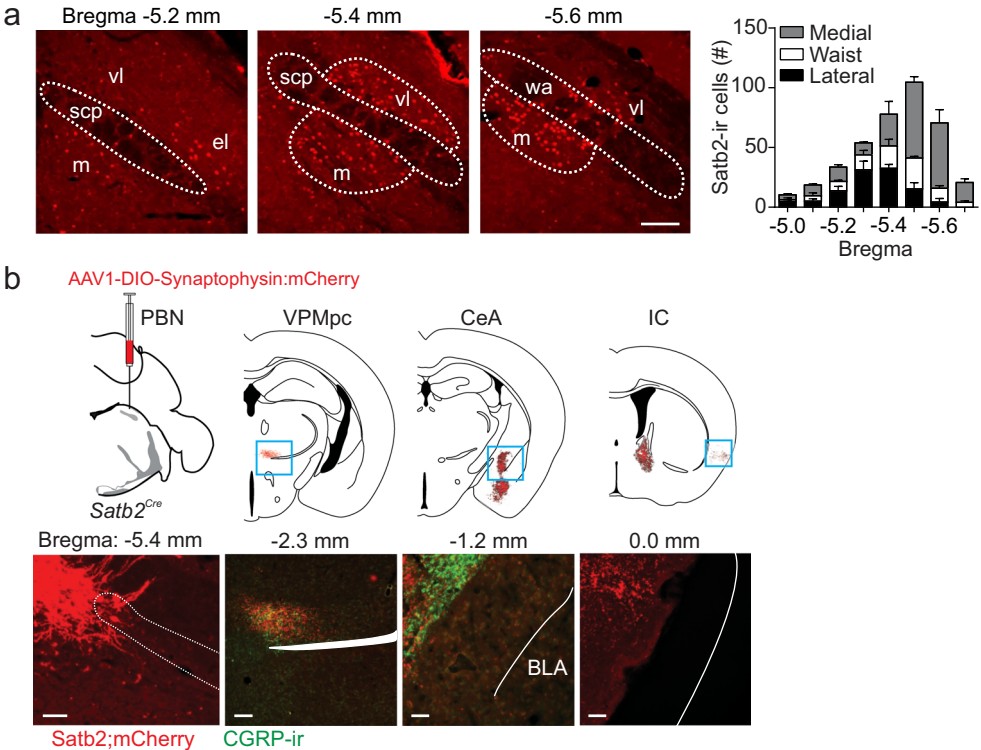

**Fig. 1 Satb2 neurons are expressed in the gustatory PBN. a** Immunostaining for Satb2 in the PBN and quantification of its expression across the rostral–caudal extent of the PBN ($n = 3$ mice). scp superior cerebellar peduncle, vl ventral–lateral, m medial, wa waist. **b** Unilateral injection of AAV1-DIO-synaptophysin:mCherry in the PBN of Satb2[Cre] mice and labeling of terminal fields in select brain regions ($n = 3$ mice). The Satb2–neuron projection overlaps with CGRP expression in the VPMpc, but not in the CeA. Scale bars, 100 µm. Data are presented as mean ± SEM. Source data are provided as a Source Data file.

monitor Satb2–neuron activity in awake animals (Fig. 2a and Supplementary Fig. 3a). We recorded neuronal responses during voluntary licking for water or different taste solutions in water-restricted mice (Fig. 2b and Supplementary Movie 1). Neurons responded reliably to the same taste stimuli across multiple trials (Fig. 2c and Supplementary Fig. 3b). Of 54 neurons recorded from three mice, 52 responded to at least one of the taste stimuli tested, including: water, 1 mM saccharin (sweet), 0.3 mM quinine (bitter), 10 mM citric acid (sour), 75 mM NaCl (salty), and 10 mM monosodium glutamate (MSG, umami) (Fig. 2b–e). These responses were sorted by the highest net average response to each stimulus and arranged in descending magnitude (Fig. 2d). Both inhibitory and excitatory responses were present, and the latency from the first lick to the peak response across all stimuli tested was $1.5 \pm 0.1$ s ($n = 210$ trials). Our results contrast with a study using fiber photometry that reported that glutamatergic neurons in the waist area of the PBN responded to all five taste modalities, but individual Satb2 neurons (measured as GCaMP6 fluorescence with a mini-microscope in awake mice) only responded to sweet tastes[31].

To identify which taste stimulus a neuron responded to best, we took the absolute value of the response to each taste during licking, normalized to the number of licks for that taste, and divided by the maximum response of that neuron (Fig. 2e). While some neurons responded best to a single taste, a large proportion of cells had similar responses to multiple tastes (Fig. 2b, e). About 20% (11/54) of neurons responded to all stimuli presented, and only a few responded to a single taste. To determine the breadth of tuning[33,34] for each neuron, we calculated the range of taste responses (entropy) using the average response during licking normalized to the number of licks for each taste (Fig. 2f). An entropy value of 1 indicates that the neuron responded to every

taste, while 0 indicates that the neuron responded to no tastes. The mean entropy for all neurons was $0.813 \pm 0.0178$. We also compared the response magnitude across tastes for each cell (noise-to-signal ratio, Fig. 2g). A noise-to-signal ratio of 1 shows that a neuron can respond equally well to two tastes, while a ratio close to 0 means a neuron responds maximally to a single taste. The mean noise-to-signal ratio was $0.601 \pm 0.077$. The correlation coefficient between the two values was 0.502 (Fig. 3h).

There was a significant increase in the magnitude of response at higher concentrations of sucrose (50 vs. 500 mM) compared to lower concentrations, along with an increase in the number of sucrose-responsive neurons when drinking the high concentration of sucrose (28 neurons for 50 mM vs. 39 neurons for 500 mM) (Supplementary Fig. 4a). No changes were apparent with different concentrations of NaCl (0.75 vs. 450 mM) or quinine (0.1 vs. 1 mM) (Supplementary Fig. 4a). Out of 43 neurons, 4/7 quinine-best and 7/18 salt-best neurons switched their best-response stimulus to sucrose at high concentrations. One quinine-best cell at low concentrations became salt-best at high concentrations (Supplementary Fig. 4b). To determine the difference in magnitude of the best response at high concentrations, we projected the average normalized response to the low concentration onto the high concentration for each taste (Supplementary Fig. 4c). A value close to 1 suggests that the neuron consistently responds best to the same taste at low and high concentrations compared to other tastes. Satb2 neurons also responded to more complex tastes such as Ensure and solid food, including both regular and sweet chow (Supplementary Fig. 4d, e). Most Satb2 neurons responded robustly to CNO-induced activation of hM3Dq (Supplementary Fig. 4f). Neurons that responded to water had comparable responses regardless of the temperature of the water (4, 22, or 40 °C), and most neurons did

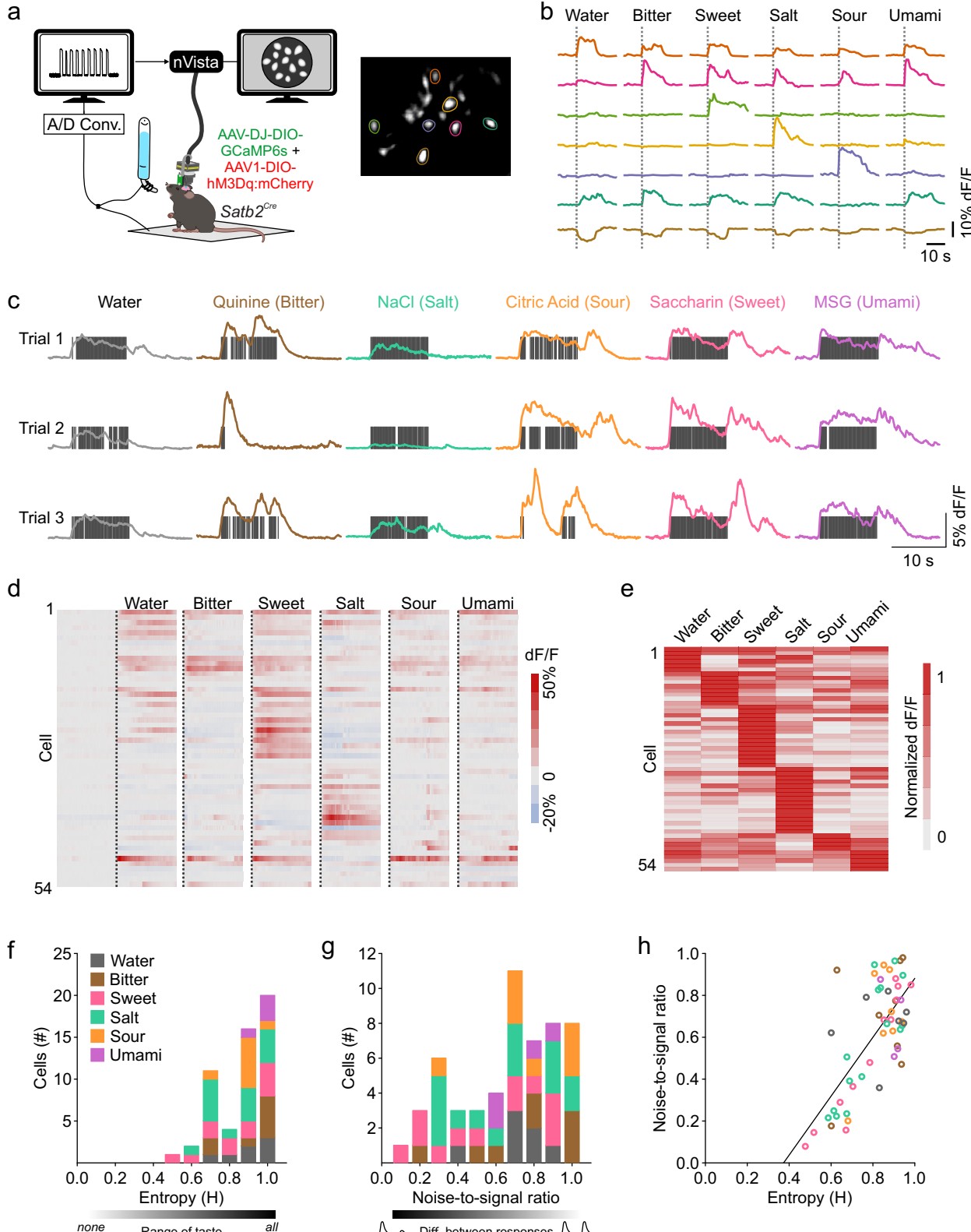

**Fig. 2 Satb2 neurons respond to taste stimuli. a** $Satb2^{Cre}$ mice were unilaterally injected with AAV-DJ-DIO-GCaMP6s and AAV1-DIO-hM3Dq:mCherry into the PBN calcium signals were visualized with an implanted GRIN lens. **b** Responses to multiple taste stimuli from cells highlighted in **a**. **c** Example responses to multiple tastes from a single neuron across three recording trials in the same day. Vertical gray bars represent licks. **d** Heat map of all 54 cells recorded showing responses of each cell during licking across six different stimuli, arranged in order of best-stimulus categories and in descending order of normalized average response magnitude to the best stimulus ($n = 3$ mice). Dotted line represents first lick. **e** Absolute value of the average response of Satb2 neurons in **d**, normalized to the number of licks. **f** Distribution of entropy values for all neurons recorded colored by their best average response. **g** Noise-to-signal ratios of responses for all neurons. **h** Correlation between entropy and noise-to-signal ratio for each neuron. Source data are provided as a Source Data file.

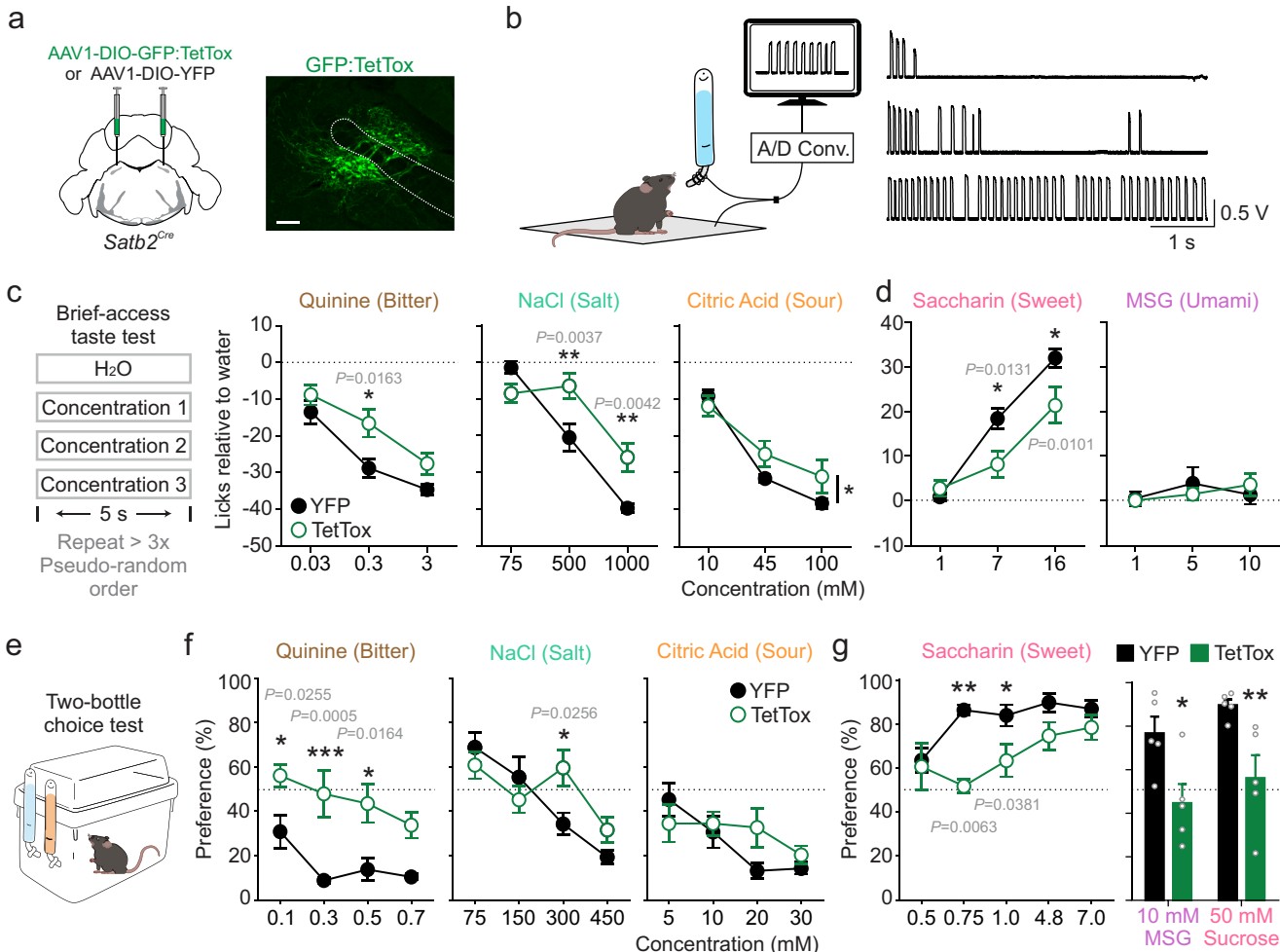

**Fig. 3 Inactivation of Satb2 neurons attenuates innate taste preferences and aversions. a** *Satb2^Cre* mice injected with AAV1-DIO-GFP:TetTox or AAV1-DIO-YFP and representative image of viral expression, similar levels of expression were verified in all TetTox mice used. Scale bar, 100 μm. **b** Mice were water restricted and given brief access to water and three concentrations of a taste solution. Example recordings of licks for different concentrations of a taste stimulus are shown. **c** In brief-access tests, TetTox mice licked more for quinine, NaCl and citric acid solutions than control mice (two-way repeated-measures (RM) ANOVA; quinine: YFP=9, TetTox $n = 5$, group effect $F_{(1,12)} = 5.658$, $P = 0.0310$; NaCl: YFP $n = 9$, TetTox $n = 6$, interaction $F_{(2,26)} = 11.6$, $P = 0.0003$; citric acid: YFP $n = 9$, TetTox $n = 6$, interaction $F_{(2,26)} = 4.806$, $P = 0.0167$). **d** TetTox mice licked less for different concentrations of saccharin but not MSG (two-way RM ANOVA; saccharin: YFP $n = 8$, TetTox $n = 6$, interaction $F_{(2,24)} = 6.586$, $P = 0.0052$; MSG: YFP $n = 9$, TetTox $n = 7$; interaction $F_{(2,28)} = 1.104$, $P = 0.3455$). **e** Mice underwent 48-h, two-bottle choice tests, with the option of water or a taste solution. **f** TetTox mice decreased avoidance of quinine and NaCl, but not citric acid (two-way ANOVA; quinine: YFP $n = 7$–14, TetTox $n = 7$–10, group effect $F_{(1,66)} = 36.96$, $P < 0.0001$; NaCl: YFP $n = 7$–13, TetTox $n = 6$–11, interaction $F_{(3,62)} = 3.378$, $P = 0.0237$; citric acid: YFP $n = 9$–14, TetTox $n = 7$–11, interaction $F_{(3,70)} = 1.673$, $P = 0.1807$). See exact n in Supplementary Table 1. **g** TetTox mice in 48-h tests also had a decreased preference for saccharin, sucrose, and MSG compared to control mice (saccharin: two-way ANOVA, YFP $n = 8$–11, TetTox $n = 4$–10, group effect $F_{(1,72)} = 17.73$, $P$; sucrose and MSG: multiple unpaired two-tailed $t$ tests, $n = 5$ mice per group; sucrose: $t(16) = 2.979$, $P = 0.0089$; MSG: $t(16) = 2.857$, $P = 0.0114$). All post hoc analyses used Holm–Sidak's multiple comparison test with *$P < 0.05$, **$P < 0.01$, and ***$P < 0.001$. Data are presented as mean ± SEM. Source data are provided as a Source Data file.

not respond to licking for an empty bottle or various odors, suggesting that Satb2 neurons do not receive substantial mechanosensory, olfactory, or thermosensory information (Supplementary Fig. 4g–j).

**Inactivation of Satb2 neurons attenuates taste responses**. To determine whether Satb2 neurons are critical for taste, we examined the contribution of Satb2 neurons to innate taste preferences. To do this, we functionally silenced Satb2 neurons in *Satb2^Cre* mice by injecting the PBN with virus carrying Cre-dependent tetanus toxin light chain (AAV1-DIO-GFP:TetTox)[35,36], while control *Satb2^Cre* mice received AAV1-DIO-YFP (Fig. 3a). We then used brief-access taste tests to examine responses to different taste stimuli. Mice were water restricted, acclimated to a custom

lickometer cage, and given 5-s access following the first lick to water and three different concentrations of a taste solution (e.g., 0.03, 0.3, and 3 mM quinine), which were pseudo-randomly presented throughout two 10-min sessions (Fig. 3b, c). Mice in both TetTox and control groups completed a similar number of trials for each taste stimulus tested (Supplementary Fig. 5a). The TetTox mice licked more for quinine, NaCl, and citric acid, suggesting that the mice found these solutions more palatable than mice with intact Satb2 neurons. However, TetTox mice decreased their licking for increasing concentrations of each aversive taste, indicating they could still distinguish between different concentrations (Fig. 3c). TetTox mice licked less for both saccharin and sucralose, indicating a decreased palatability of these sweet solutions (Fig. 3d and Supplementary Fig. 5b). Neither group showed a clear preference or aversion for MSG (Fig. 3d).

To further examine changes in taste preferences, mice were given 48-h, two-bottle choice tests between water and a taste solution (Fig. 3e). Compared to control mice, TetTox mice showed an increased preference for multiple concentrations of quinine and an aversive concentration of NaCl, but there were no differences in preference for citric acid at any concentration (Fig. 3f). TetTox mice showed a decreased preference for 4.8 mM

saccharin compared to control mice, but not at 7 mM (Fig. 3g); however, TetTox mice drank significantly lower volumes of saccharin at higher concentrations, suggesting that they found saccharin less appetitive than control mice (Supplementary Fig. 5c). Similarly, TetTox mice decreased their preference for 10 mM MSG and 50 mM sucrose (Fig. 3g). Taken together, inactivation of Satb2 neurons interfered with innate taste

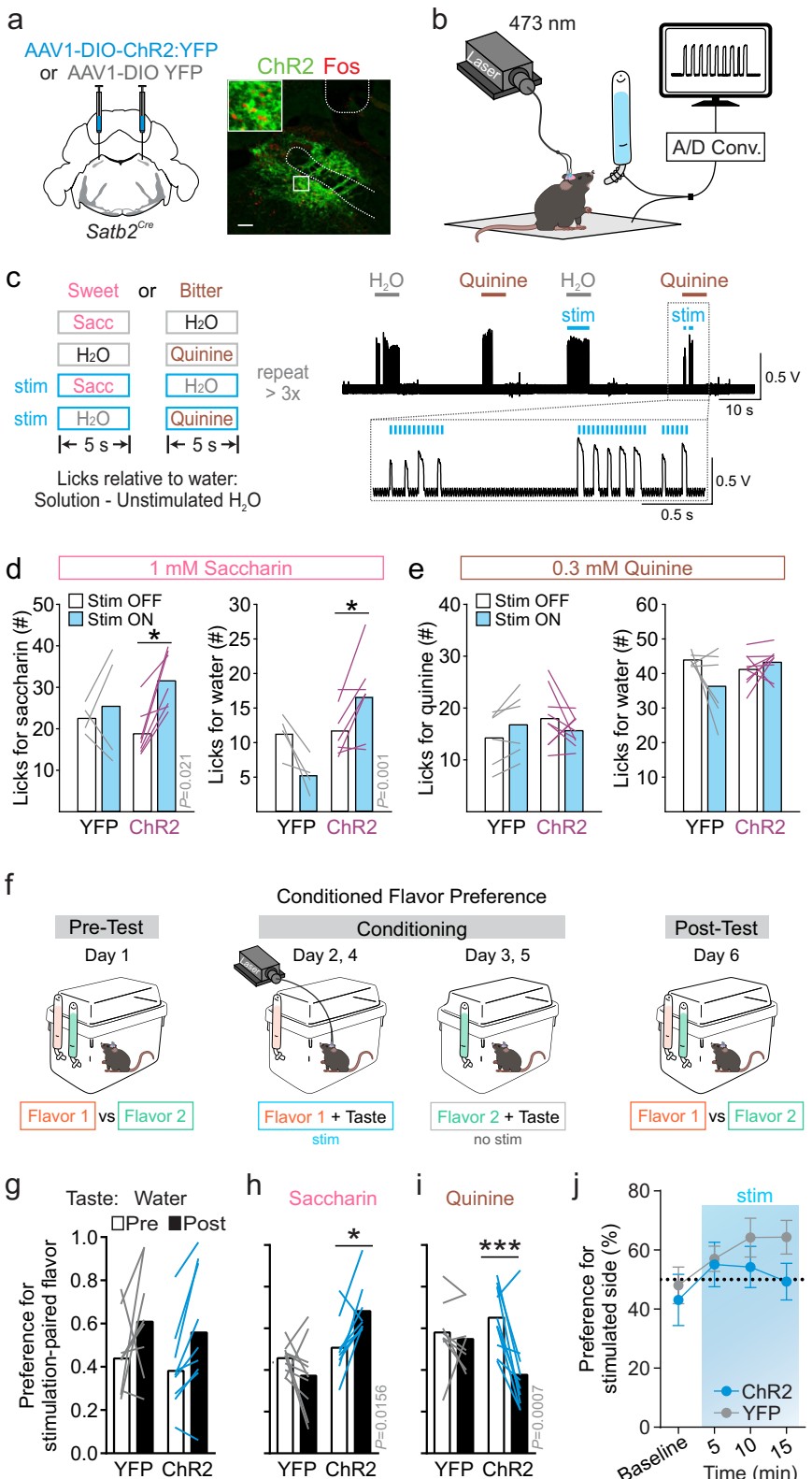

**Fig. 4 Activation of Satb2 neurons enhances taste perception. a** Bilateral injections of AAV1-DIO-ChR2:YFP or AAV1-DIO-YFP into the PBN of *Satb2^Cre^* mice and representative image of viral expression, cannula placement, and Fos induction following photostimulation. Similar levels of viral expression and cannula placement were verified for all mice shown. Scale bar, 100 μm. **b** Diagram of custom lickometer setup. **c** Brief-access, taste-test paradigm with lick-triggered 30-Hz photostimulation and representative recording from one trial. **d** Satb2 photostimulation caused ChR2 but not YFP-expressing mice to lick more for the paired 1 mM saccharin solution (YFP $n = 4$, ChR2 $n = 7$; one-way RM ANOVA, interaction $F_{(3,12)} = 4.512$, $P = 0.024$). Satb2 stimulation also increased licking for water in ChR2 mice during these trials (one-way RM ANOVA, interaction $F_{(3,12)} = 9.133$, $P = 0.002$). **e** Satb2 stimulation had no effect on licking for water or 0.3 mM quinine paired with Satb2 stimulation (YFP $n = 6$, ChR2 $n = 9$; one-way RM ANOVA, quinine: interaction $F_{(3,16)} = 2.174$, $P = 0.131$; water: interaction $F_{(3,16)} = 2.150$, $P = 0.134$). **f** Illustration of conditioned flavor preference experiment. **g** Neither YFP nor ChR2-expressing mice develop a preference for flavored water paired with PBN photostimulation (YFP $n = 8$, ChR2 $n = 9$; two-way RM ANOVA, $F_{(1,15)} = 0.0035$, $P = 0.9534$). **h** If 1 mM saccharin was added to flavors during conditioning, activation of Satb2 neurons in ChR2-injected mice resulted in a preference for the flavor paired with photostimulation (YFP $n = 10$, ChR2 $n = 9$, two-way RM ANOVA, $F_{(1,17)} = 10.42$, interaction $P = 0.0049$). **i** If 0.1 mM quinine was added to flavors during conditioning, activation of Satb2 neurons in ChR2-injected mice resulted in an aversion for the flavor paired with photostimulation (YFP $n = 8$, ChR2 $n = 10$, two-way RM ANOVA, interaction $F_{(1,16)} = 7.133$, $P = 0.0167$). **j** Photoactivation of Satb2 neurons had no effect in a real-time, place-preference test ($n = 10$ mice per group; two-way RM ANOVA, interaction $F_{(3, 54)} = 1.0$, $P = 0.3998$). Post hoc analyses were done with Turkey's or Holm–Sidak's multiple comparison test with *$P < 0.05$, **$P < 0.01$, ***$P < 0.001$, and ****$P < 0.0001$. Data are presented as mean ± SEM. Source data are provided as a Source Data file.

preferences or aversions for most tastes tested and brought them closer to a neutral value (0 licks relative to water in the brief-access test or 50% preference in the two-bottle tests).

Inactivation of Satb2 neurons had no impact on food or water intake either at baseline or following deprivation (Supplementary Fig. 5d–f). Mice maintained similar body weights and showed no changes in motivation to lever press for food or sucrose pellets on a progressive-ratio task (Supplementary Fig. 5g–i). In addition, there was no difference in the latency to first press for sucrose pellets between the groups, although TetTox mice initially discarded more sucrose pellets than controls (Supplementary Fig. 5j, k). These data suggest that Satb2 neurons contribute to taste perception and likely food choice, but they do not play a role in the motivation to eat and drink.

**Activation of Satb2 neurons enhances taste responses.** We next examined whether activation of Satb2 neurons could alter the perception of sweet and bitter tastes. We bilaterally injected a virus to induce Cre-dependent expression of channelrhodopsin (AAV1-DIO-ChR2:YFP) or a control virus (AAV1-DIO-YFP) and placed fiber-optic cannulae over the PBN of *Satb2^Cre^* mice (Fig. 4a and Supplementary Fig. 6a). Mice were water restricted and placed in a custom lickometer cage where they were given 5-s access to a bottle containing water or 1 mM saccharin, with or without lick-triggered 30-Hz optical stimulation (Fig. 4b, c). ChR2 (channelrhodopsin-2)-expressing mice licked more for saccharin that was paired with photostimulation than saccharin without stimulation, whereas control mice showed no differences in intake between the two solutions (Fig. 4d). Mice also showed a preference for the stimulation-paired water solution, in agreement with previous results[31], suggesting that Satb2 neuronal stimulation increased the palatability of both saccharin and water. In a similar paradigm, photostimulation of Satb2 neurons did not alter licking for quinine or water. As mice appeared to be at relative licking ceilings for both water and quinine, we decreased the concentration of quinine and again saw no effect (Supplementary Fig. 6b). Mice in both ChR2 and control groups completed a similar number of trials for each taste stimulus tested (Supplementary Fig. 6c). These data suggest that activation of Satb2 neurons enhances the perceived palatability of a sweet taste and water, whereas activation did not change perception of a bitter solution.

We also tested whether stimulation of Satb2 neurons could alter a taste-associated memory in a flavor-paired conditioning paradigm. First, water-deprived mice were presented with two bottles of flavored water to determine their initial preference for the flavors. Mice then underwent conditioning trials where they were given 10-min access to a single flavor in water each day, where one flavor was paired with 30-Hz photostimulation and

one was not. Conditioning occurred twice for each solution. The next day, mice underwent a two-bottle choice test for the two flavors (Fig. 4f). There was no change in preference for the stimulation-paired flavor, with no added tastant, following this paradigm (Fig. 4g). However, if during conditioning both flavors were presented in a 0.75 mM saccharin solution instead of water, then ChR2-stimulated mice showed an increased preference for the stimulation-paired flavor during the post-conditioning, two-bottle test (Fig. 4h). Mice showed a decreased preference for the stimulation-paired flavor if the flavors were presented in a quinine solution during conditioning (Fig. 4i). However, mice did not develop a preference or aversion for the photostimulation-paired chamber during a real-time, place-preference test (Fig. 4j). Given that subpopulations of Satb2 neurons can be differentially activated or inhibited by the five basic tastes, activating the entire population is likely to be a non-physiological manipulation that creates competing taste sensations. However, collectively increasing the activity of Satb2 neurons can alter the perception of both bitter and sweet tastes.

**CGRP neurons in the PBN contribute to aversive taste processing.** Satb2-expressing neurons are unlikely to represent all taste-responsive neurons in the PBN because studies have identified bitter-responsive neurons in the external lateral PBN[1,37], an area with minimal Satb2. However, this region is where neurons that express calcitonin gene-related peptide (CGRP) reside, which are critical for developing and maintaining a CTA[23,24]. To determine whether these CGRP neurons are important for innate taste preferences, we functionally silenced CGRP neurons by injecting AAV1-DIO-GFP:TetTox, or AAV1-DIO-YFP as control, into the PBN of *Calca^Cre^* mice, which encodes CGRP (Fig. 5a)[28]. Inactivation of CGRP neurons led to an increased preference for quinine and an aversive concentration of salt compared to control mice, although there were no differences in preference for citric acid or saccharin (Fig. 5b). Inactivation of CGRP neurons did not affect 12-h water intake (Fig. 5c).

To verify that Satb2 and CGRP label distinct populations of neurons, we generated a mouse line that has tdTomato (tdT) targeted to the *Calca* gene (Supplementary Fig. 7a, b). Immunostaining for Satb2 in *Calca^tdT^* mice showed minimal overlap between the two markers (Fig. 5d, e). As inactivation of Satb2 or CGRP neurons decreased avoidance of aversive tastes, we tested whether simultaneous inactivation of both populations would have an additive effect. To do this, we silenced CGRP neurons alone using *Calca^Cre^* or both CGRP and Satb2 neurons using *Satb2^Cre^*::*Calca^Cre^* double transgenic mice (Fig. 5f). In brief-access tests, *Calca^Cre^* mice expressing TetTox increased licking for quinine but not for citric acid solutions compared to YFP-

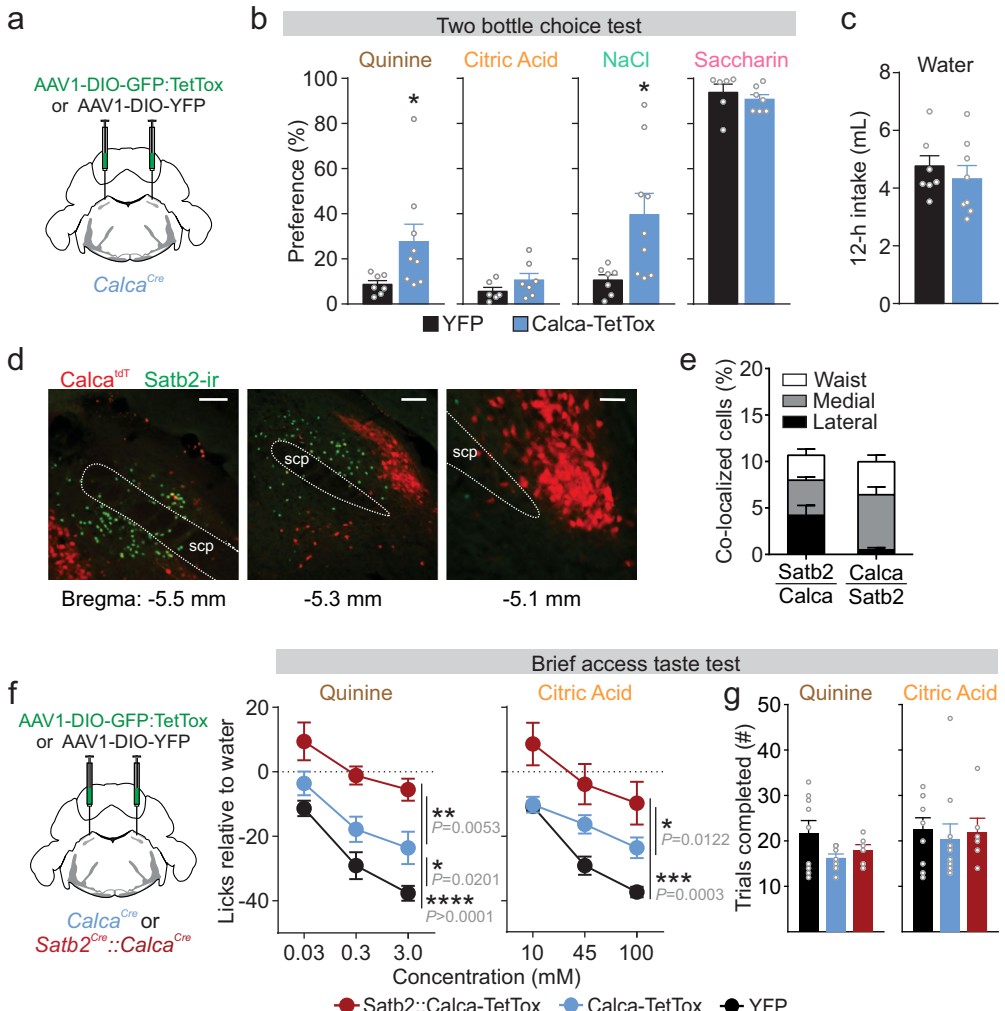

**Fig. 5 CGRP neurons contribute to the avoidance of aversive tastes. a** *Calca*$^{Cre}$ mice were injected bilaterally with either AAV1-DIO-GFP:TetTox or AAV1-DIO-YFP. **b** In 48-h, two-bottle choice tests, Calca-TetTox mice decreased avoidance of 0.3 mM quinine and 300 mM NaCl, but there was no change in their preference for 30 mM citric acid or 4.8 mM saccharin (two-tailed unpaired *t* test with Welch's correction, YFP *n* = 5, TetTox *n* = 9. Quinine: *t*(8.737) = 2.384, *P* = 0.0417. NaCl: *t*(8.969) = 2.95, *P* = 0.0163, two-tailed unpaired *t* test, YFP *n* = 6, Tetox *n* = 7. Citric acid: *t*(11) = 1.405, *P* = 0.1876. Saccharin: *t*(11) = 0.7602, *P* = 0.4631.) **c** There was no difference in overnight water intake (*n* = 8 per group, two-tailed unpaired *t* test, *t*(14) = 0.7712, *P* = 0.4534). **d** tdTomato expression in the PBN of a *Calca*$^{tdT}$ mouse (red), with immunostaining showing minimal colocalization with Satb2 (green). Scale bars, 100 μm. scp superior cerebellar peduncle. **e** Quantification of overlap between tdTomato in *Calca*$^{tdt}$ mice and immunolabeled Satb2 neurons in the PBN. **f** In brief-access taste tests, mice with TetTox in both Satb2 and CGRP neurons were unable to detect aversive tastes (two-way RM ANOVA; quinine: YFP *n* = 9, Calca *n* = 9, Satb2::Calca *n* = 7, group *F*$_{(2,22)}$ = 16.42, *P* = 4.33E − 05; citric acid: YFP *n* = 9, Calca *n* = 10, Satb2::Calca *n* = 7, group *F*$_{(2,23)}$ = 11.21, *P* = 0.0004. **g** No difference in number of trials completed between TetTox and control mice (Kruskal–Wallis, quinine: *H*$_{(2)}$ = 1.393, *P* = 0.498; Kruskal–Wallis, citric acid: *H*$_{(2)}$ = 1.116, *P* = 0.572) Post hoc analyses were done with Holm–Sidak's multiple comparison test with \**P* < 0.05, \*\**P* < 0.01, \*\*\**P* < 0.001, \*\*\*\**P* < 0.0001. Data are presented as mean ± SEM. Source data are provided as a Source Data file.

expressing controls (Fig. 5f). Simultaneous inactivation of both Satb2 and CGRP neurons resulted in mice that licked for quinine and citric acid at the same rate as water (Fig. 5f). Mice completed a similar number of taste trials (Fig. 5f). Inactivation of CGRP neurons had no effect on licking for 1, 7, or 16 mM saccharin solutions relative to control mice (*n* = 6 mice per group, two-way RM ANOVA; group *F*$_{(1,10)}$ = 0.2488, *P* = 0.6287). These data suggest that CGRP neurons relay information about aversive taste that is complementary to, but distinct from, Satb2 neurons.

## Discussion

Using calcium imaging in awake, freely licking mice, we revealed that Satb2 neurons in the PBN can rapidly respond to water and all taste stimuli, but we did not detect responses to

mechanosensory stimulation, temperature, or odors. Inactivation of these neurons caused deficits in their preference for sweet, salty, and bitter solutions in both brief- and long-access taste tests. Activation of Satb2 neurons enhanced palatability of sweet but not bitter tastes and increased the innate preference or aversion for both tastes in a conditioning paradigm. We also show that CGRP neurons in the PBN contribute to the perception of aversive tastes, and inactivation of both Satb2 and CGRP neurons in the same mice eliminated aversions to bitter and sour tastes altogether. Together, these data demonstrate that taste information diverges and is relayed by parallel populations of neurons as early as the secondary nucleus in the taste circuitry.

Subsets of Satb2 neurons responded to all taste qualities presented, and many of these neurons were broadly tuned and responded robustly to multiple tastes. Studies profiling

taste-responsive neurons in the PBN have reported similar results, although most describe a greater number of narrowly tuned neurons[12,15,37–39]. Many prior studies compared taste responses to that of water; hence, they did not report responses to water. Given that there is evidence that water can be detected by distinct mechanisms[40], we designed our experiments to independently analyze response to water and taste stimuli and identified a subset of water-best neurons. While some taste-sensitive neurons respond to temperature and mechanosensation[41,42], this was not apparent in Satb2 neurons that we recorded. These differences could be in part due to differences in recording from anesthetized and awake mice, and additional profiling of Satb2 vs. other neuronal populations within the PBN is needed. In addition, MSG and 1 mM saccharin are considered weak or peri-threshold umami and sweet stimuli, respectively[43–45]. Alternative compounds or concentrations could reveal greater neuronal responses.

Subsequent experiments comparing neuronal responses to low and high concentrations of taste solutions revealed that high concentrations of sweet, but not salty or bitter, recruited a greater number of neurons and changed the overall response profile of Satb2 neurons. About half of neurons recorded switched to a sweet-best response, which is in agreement with a previous study reporting that Satb2 neurons preferentially respond to sweet[31]. However, unlike that study, we found that most Satb2 neurons are broadly tuned and subsets of neurons respond well to tastes other than sweet, even at a saturating concentration of sucrose. This discrepancy could be due to in part to subtle anatomical differences in recording sites. We recorded from the waist and medial PBN, with the lens located over the more caudal half of Satb2 neurons, which is similar but perhaps slightly rostral to the location described by Fu et al.[31]. Overall, because many Satb2 neurons responded to multiple tastes, our data do not support the idea that they represent labeled lines in taste processing[2,3]. While parabrachial Satb2 neurons appear to be broadly tuned, there could be other neurons in the PBN that respond to discrete tastes.

In agreement with our in vivo imaging data, manipulation of Satb2 neurons had a large effect on the ability of mice to respond appropriately to multiple tastes. Inactivation experiments blunted innate preferences for all tastes, with the most significant effects on bitter and sweet tastes. However, mice were still able to detect and respond to all solutions tested, although this could be in part due to incomplete coverage of all Satb2 neurons with viral transductions. We were able to enhance aversion to bitter and increase the palatability of a sweet solution by photostimulating Satb2 neurons. This bidirectional effect is notable because in both cases the majority of Satb2 neurons were activated, illustrating that Satb2 neurons have a complex role in taste that is not as simple as encoding a single taste or value. Given the large behavioral effects we observed on bitter taste, it is surprising that we did not identify a more robust population of bitter-responsive neurons; perhaps, there are more bitter-responsive Satb2 neurons in areas rostral or lateral areas that we did not image, in agreement with reports indicating that bitter-best neurons are concentrated in the lateral PBN[15]. Collectively, our data support the idea that Satb2 neurons are important for normal taste perception.

Satb2 is not a comprehensive marker for taste-associated neurons in the PBN, as there are subnuclei containing taste-responsive neurons that do not express Satb2[31,37,46]. Here we identify a role for CGRP neurons by showing that inactivation of CGRP neurons causes animals to be more accepting of bitter and aversive salt solutions. While there were minimal or no effects on sour avoidance after the loss of either CGRP or Satb2 neurons, there was a surprising additive effect of inactivating both CGRP and Satb2 neurons that caused animals to lick for sour solutions at about the same rate as water. This result implies that these two populations together are necessary to detect sour, but each can compensate for the loss of the other. Similarly, inactivation of both CGRP and Satb2 had an additive effect on licking for bitter solutions, where again consumption of the bitter solution became comparable to water even at high concentrations. We predict a similar effect for aversive concentrations of salt given that inactivation of CGRP or Satb2 neurons alone decreased salt avoidance. There was no indication that CGRP neurons are needed for perception of sweet, which is unsurprising given that CGRP neurons are thought to function as a house alarm and encode diverse aversive stimuli such as foot shock, itch, and malaise[28,35,47]. Collectively, these data suggest that CGRP neurons relay the negative valence of aversive tastes, whereas Satb2 neurons likely contribute to taste identity.

Satb2 was selected as a neuronal marker for this study due to its restricted expression pattern in the gustatory PBN. The role of Satb2 itself is unclear but could be developmental, as it is a transcription factor needed for proper gene expression and connectivity in the cortex[48]. Satb2 neurons in the PBN are well positioned to be part of the taste circuitry; they respond to taste and send strong projections to areas that have been associated with both the identity and valence of taste, including the VPMpc, CeA, and BNST[6,16,21]. CGRP neurons project to many of these same brain regions[24,28]. Satb2 neurons also send a strong projection to the BMA, which is atypical for most PBN neurons[49,50], suggesting that this projection may have a special function. Functionally, Fu et al.[31] has demonstrated that optogenetic activation of Satb2 terminals in the VPMpc (but not BNST or CeA) increased the palatability of water. Additional exploration of brain regions innervated by Satb2 neurons will be important to reveal their functional significance and expand our understanding of taste circuitry.

Given the complex role of the PBN in gustatory, visceral, pain, and thermal processing, discovering a way to target functionally and anatomically distinct neuronal populations is an important step in dissecting various inputs involved in sensory processing. Satb2 is the first genetic marker to be identified for a population of taste-responsive neurons in the PBN that is providing a foothold to further characterize taste neurons and their function both in the PBN and other connected nodes of the taste circuitry.

## Methods

All experiments were approved by the Institutional Animals Care and Use Committee at the University of Washington. Heterozygous $Calca^{[Cre28}$, $Calca^{tdt}$, heterozygous and homozygous $Satb2^{Cre}$ or C57Bl/6 J (wild-type) mice were used for all experiments, ranging in age from 6 to 24 weeks. Male and female mice from the same litter were split randomly between control and experimental groups, roughly half and half per group. No formal comparisons were done comparing the responses of male and female mice. Animals were group housed with littermates before and after stereotaxic surgery and singly housed prior to the start of behavior experiments. During experiments, mice were individually housed on a 12-h cycle (lights off 17:00–5:00) at ~22 °C with ad libitum access to water and food (Picolab, #5053), unless otherwise indicated in experiments below.

**Generation of $Satb2^{Cre}$ mouse line**. As described previously[51], the 5′ arm (~ 5.2 kb with $Pac1$ and $Sal1$ sites at each end) and 3′ arm (~5.2 kb with $Pme1$ and $Not1$ sites at each end) of $Satb2$ gene were amplified from a C57BL/6 BAC clone by PCR using Phusion Polymerase (New England Biolabs, Ipswich, MA) and cloned into polylinkers of a targeting construct that contained IRES-mnCre:GFP, an frt-flanked Sv-Neo gene for positive selection, and HSV thymidine kinase and $Pgk$-diphtheria toxin A chain genes for negative selection. The IRES-mnCre:GFP cassette has an internal ribosome entry site followed by a Myc-tag and nuclear localization signal at the N terminus of Cre recombinase fused to green fluorescent protein. The construct was electroporated into G4 ES cells (C57Bl/6 × 129Sv hybrid) and correct targeting was determined by Southern blot of DNA digested with $Bgl$I using a $^{32}$P-labeled probe downstream of the 3′ arm of the targeting construct. Three of 83 clones analyzed were correctly targeted. One clone that was injected into blastocysts resulted in good chimeras, which transmitted the targeted allele throughout the germline. Progeny were bred with $Rosa26$-$FLP$ recombinase to remove the SV-Neo gene. Mice were then continuously backcrossed to C57Bl/6 mice.

**Generation of *Calca*^tdT mouse line**. A cassette including the tdTomato open-reading frame was inserted into the *Calca* locus between a *Sal*1 site that had been engineered just 5′ of the initiation codon in exon 2 and an *Eco*N1 site in non-coding exon 6. An frt-flanked Sv-Neo gene was also included for positive selection. The targeting construct had 6.7 kb of 5′ flanking sequence and 4.2 kb of 3′ flanking sequence. *Pgk*-DTa and HSV-TK genes were also included for negative selection. The construct was electroporated into G4 (129/Sv × C57Bl/6) ES cells and correct targeting was determined by Southern blot of DNA cut with *Stu*1 using a ^32P-labeled probe located beyond the end of the 5′ arm. Fifty out of 82 clones analyzed were corrected targeted. Cells from several clones were injected into C57Bl/6 blastocysts and one of these resulted high-percentage chimeric mice. Chimeras were bred with *Gt(Rosa26)Sor*-FLPase mice to remove the Sv-Neo gene. Mice were subsequently backcrossed to C57Bl/6 mice.

**Viruses**. AAV serotype 1 viruses (AAV1-Ef1α-DIO-Synaptophysin:mCherry, AAV1-Synapsin-DIO-YFP, AAV1-Ef1α-DIO-mCherry, AAV1-DIO-hSyn-hM3Dq:mCherry, AAV1-CBA-DIO-GFP:TetTox) were generated at the University of Washington as described[52]. AAV-DJ-EF1α-DIO-GCaMP6s was produced by the UNC Vector Core and provided by G. Stuber. All viruses were injected at a titer of ~10^9 viral particles/μL.

**Stereotaxic surgery**. Mice (6–9 weeks old) were anesthetized with isofluorane and placed in a small-animal stereotax (Kopf Instruments). Coordinates for the anterior-posterior plane were normalized using a correction factor (F = Bregma-Lambda distance/4.21). Cre-dependent virus was bilaterally injected into the PBN (antero-posterior (AP), −5.0 mm; medio-lateral (ML), ±1.25 mm; dorso-ventral (DV), −3.3 mm) or IC ((AP, 1.54; ML, 3.5; DV, 3.0) and (AP, 1.0; ML, 3.7; DV, 3.6)) over 10 min for a final volume of 0.3 μL using glass capillaries and pressure injection (Nanoinject II, Drummond Scientific). Mice were given a minimum of 2 weeks of recovery before starting experimental manipulations. Animals that received viral injections that missed the target (<10 fluorescent cells visible/section) were excluded from analysis.

**Histology**. Mice were anesthetized with phenytoin/pentobarbital and perfused with phosphate-buffered saline (PBS; pH 7.4), followed by 4% paraformaldehyde (PFA) in 0.1 M phosphate buffer (PB; pH = 7.4). After an overnight post-fix in 4% PFA, brains were cryoprotected overnight in 30% sucrose before being embedded in OCT and stored at −80 °C. Free-floating sections (30 μm) were prepared with a cryostat. They underwent three washes in PBST (PBS + 0.3% Triton X-100) and were blocked in 3% normal donkey serum in PBST (1–2 h). Sections were then incubated overnight at 4 °C in primary antibodies, including rabbit anti-Satb2 1:2500 (ab34735, Abcam), mouse anti-Satb2 1:1500 (ab34735, Abcam), rabbit anti-Fos 1:2000 (#22505, Cell Signaling; ab190289, Abcam), goat anti-Fos 1:300 (sc-52-G, Santa Cruz), chicken anti-GFP 1:10,000 (ab13970, Abcam), goat anti-CGRP 1:1000 (ab36001, Abcam), and/or rabbit anti-dsRed 1:1000 (632475, Takara). The next day, the tissue was washed three times in PBS and incubated 1–2 h in appropriate secondary antibodies including Alexa Fluor 488/594 donkey anti-sheep, Alexa Fluor 488/594 donkey anti-goat, Alexa Fluor 488/594 donkey anti-rabbit, or Alexa Fluor 488 donkey anti-chicken (1:500; Jackson Immunoresearch Laboratories). Tissue was washed three times in PBS, mounted onto glass slides, and coverslipped with Fluoromount-G (Southern Biotech).

**Calcium imaging**. Three weeks after viral injection with AAV with Cre-dependent GCaMP6s and hM3Dq, mice were anesthetized with isoflurane and implanted with a GRIN lens (6.1 mm length, 0.5 mm diameter; Inscopix #1050-002211) with the assistance of a ProView implant kit (Inscopix, #100-000754) and injection of CNO (1 mg/kg) that allowed visualization of fluorescent activity during implantation. The lens was targeted to be ~200–300 μm above the neurons using the following coordinates: AP −5.0 mm, ML −1.2 mm, and DV 2.8 mm. Two weeks after lens implantation, mice were anesthetized and a baseplate (Inscopix, #100-000279) was implanted above the lens. The baseplate provides an interface for attaching the miniature microscope during calcium-imaging experiments, but at other times a baseplate cover (Inscopix, #100-000241) was attached to prevent damage to the lens. Calcium fluorescence was recorded at 10 frames/s with nVista 2 software (Inscopix). We initially used the same AAV1-CBA-DIO-GCaMP6m virus that was used successfully for expression in *Calca*^Cre mice;[24,47] however, we never visualized fluorescence during lens placement even with CNO treatment to activate hM3Dq in the Satb2 neurons. Switching virus to AAV-DJ-EF1α-DIO-GCaMP6s allowed visualization of green fluorescent cells, but we have not determined whether the different serotype or promoter is responsible. Out of 30 mice injected with GCaMP6s, we observed fluorescent activity in eight mice during lens implantation. Cells were only apparent in four mice upon installation of the baseplate, one of which only had two cells visible and was not used for experiments.

During taste exposure, lick patterns were recorded with a custom-made lickometer setup[53] and temporally synchronized to calcium recordings with a TTL trigger. Recording sessions consisted of a 30-s baseline, 10-s access to water or taste starting from the first lick, followed by a 60-s post-lick period with a 2-min inter-trial interval between recordings. Each session was repeated three times with >1 h between sessions. The same procedure was used for recordings during exposures to

different taste concentrations, vanilla Ensure, water temperatures, and an empty bottle. All solutions unless otherwise specified were presented at room temperature.

For responses to odor, mice were placed in a clean cage and baseline activity was recorded for 30 s prior to presentation of odor on a cotton-tipped applicator dipped in extracts of almond, vanilla, raspberry, and orange, or fox urine (Harmon) and 50 mM citric acid. Mice approached the stimulus at least three times for each odor. For food consumption, mice were fasted overnight and placed in a clean cage. Baseline activity was recorded for 60 s prior to presentation of a food pellet and mice consumed the food for 2 min. All behaviors were recorded with a video camera (Logitech) synchronized to calcium fluorescence recordings.

**Image processing**. Raw images were pre-processed, spatially down-sampled (4-pixel bins), and motion-corrected using Inscopix Data Processing Software (v1.2, Inscopix). As many Satb2 neurons exhibit low baseline calcium fluorescence, regions of interest (ROIs) were manually identified using images acquired during sucrose consumption or after artificial activation of Satb2 neurons with CNO. ROIs were semi-automatically aligned across imaging sessions by shifting the ROI positions drawn previously in the *x* and *y* direction to maximize pixel intensity and manually corrected as necessary. d*F*/F was calculated using the formula $(F − F_0)/F_0$, where $F_0$ is the average fluorescence during the 30 s before first lick or stimulus.

To categorize Satb2 neurons by their best-response taste stimulus, the net response was calculated by subtracting the average 30-s baseline prior to first lick from the average response during licking. Cells were arranged in order of highest net positive response to a stimulus and further sorted in descending response magnitude to that stimulus. Neurons were deemed taste-responsive if d*F*/F > 3 SD within 2 s of stimulus onset (or first lick) and persists for >5 frames. Further analysis was done by summing the absolute value of the responses for each neuron during licking and dividing by the number of licks to account for potential differences in lick rate. To facilitate comparisons across neurons, the responses for each cell were then normalized to the maximum response for each experiment. The range of taste response sensitivity for each neuron was calculated using the formula for entropy[34]

$$H = -K \left( \sum_{i=1}^{6} Pi \log Pi \right), \qquad (1)$$

where Pi represents the proportional response to each of the six stimuli used and K is a scaling constant that limits $0 > H \le 1$ (1.2851 for six stimuli). Neurons that respond to many stimuli (broadly tuned) have higher H values, while H = 0 for neurons that respond to only one stimulus (narrowly tuned). The noise to signal[33] was also calculated by dividing the response to the second-best stimulus (noise) by the response to the best stimulus (signal) with $0 > \text{ratio} \le 1$. A neuron that responded to only one stimulus will have a ratio = 0, while a neuron that responded equally to two stimuli will have a ratio = 1.

**Brief-access taste tests**. Mice were maintained on a water-restriction schedule, and testing was performed in two 10-min sessions, one in the morning and one in the afternoon. Mice were returned to their home cage between the morning and afternoon sessions and given 2-h water access following the afternoon session. Tests were done in custom lickometer cages[53] with two water ports available in the front of the cage and a floor made of aluminum foil or a wire-grid attached the grounded housing (shield) of a BNC input connector of a Digidata 1440A analog–digital converter (Molecular Devices). The metal sipper tube on bottles for taste solutions was attached to the central pin of the BNC connector. A positive voltage step was recorded whenever the mouse contacted the metal sipper tube, and <60 nA was passed through the tongue with each lick. Licking behavior was recorded and analyzed on the pClamp 11.0 software (Molecular Devices).

Mice were acclimated to the water-restriction schedule and test cages for 2 days, and on day 3, they were trained to lick water presented in metal sipper tubes in 5-s trials. Testing occurred on days 4–6. Mice were given water or a taste solution. Each trial lasted for 5 s, beginning from the time of first lick, with an ~10 s inter-trial interval. Solutions were presented in a pseudorandom order arranged into blocks of 4 (water and three concentrations of taste solution), which included quinine (0, 0.03, 0.3, and 3 mM), citric acid (0, 10, 45, and 100 mM), and NaCl (0, 75, 500, and 1000 mM). After 24-h recovery, mice were mildly food and water restricted by giving them 1.5–2 g of both food and water at the start of the dark cycle. On the subsequent day, mice underwent testing for either saccharin (0, 1, 7, and 16 mM), sucralose (0, 1, 5, and 10 mM), or MSG (0, 1, 5, and 10 mM). Mice were given 5-s access to each solution during each trial starting from the first lick. Mice were required to sample water and all three concentrations of the taste solution at least three times to be included in the analysis. All solutions were at room temperature.

**Flavor-paired conditioning**. Mice were maintained on a water-restriction schedule where testing was performed during a 10–30 min period in the morning and there was 2-h water access in the afternoon. All tests were performed in the home cage. Mice were acclimated to the water-restriction schedule for at least 3 days. For the pre-test, mice were given 30-min access to two bottles with distilled water flavored with 0.1% orange (Kroger) and 0.25% almond (Kroger), 0.075% lemon-lime (SodaStream), and 0.1% raspberry (Frontier Co-Op), or 0.15% vanilla (Kroger) and 0.2% raspberry. On days 2–5, mice underwent 10-min conditioning trials, where

they were given single-bottle access to one of the two flavors on alternative days. One flavor was paired with lick-triggered photostimulation (30 Hz, 10 ms). On the final test day, mice were again given a 30-min, two-bottle choice test for the flavors in distilled water. For taste conditioning, the flavors contained either 0.75 mM saccharin or 1 mM quinine on days 2–5. In all tests, the pre and post two-bottle choice tests were flavors in distilled water. The flavor paired with stimulation was counterbalanced between cohorts of mice. The same mice underwent two of the conditioning paradigms, with different flavors used for each test.

**Food and water intake**. Mice were housed in BioDAQ recording chambers to monitor food and water intake in real time (ResearchDiets). Mice were given a minimum of 3 days to acclimate to the cages. Baseline intakes of food and water are the average of 3 days after the acclimation period. For deprivation experiments, water or food was removed at the start of the dark cycle and returned the following morning around 09:00 and intake was recorded for 2 h.

**Two-bottle choice preference test**. Mice were acclimated in a home cage with two water bottles held in ports in the cage wall for a minimum of 3 days. The 48-h taste tests were performed with a taste solution and distilled water. The taste solutions were given to the animals in a pseudorandom order so that each animal had a different taste history. Intake was monitored daily and the average over 2 days was used to calculate preference. The side with the taste solution was switched daily to avoid a side preference.

**Operant conditioning**. YFP and TetTox mice were food deprived to 90% of their body weight and trained to lever press for food (Bio-Serv Precision Pellets, F0071) on a fixed ratio training protocol. Only one lever was extended, and this lever was consistent between days for each animal, but random between animals. Mice were trained in the morning and fed directly after training. After 3 consecutive days with over 20 presses in a 1-h trial, animals were moved to a progressive-ratio task with an active and inactive lever. The location of active lever was switched from the previous task. The task concluded when there was less than a 10% change in break point for three consecutive days. Mice were returned to ad libitum feeding schedule and allowed to press for sugar pellets (Bio-Serv, F05550) on consecutive mornings. The first day of testing, the morning after returning to an ad libitum diet, was excluded from analyses. Animals were tested daily for 2 weeks and break point, latency to first lever press, and a number of pellets not consumed were recorded.

**Quantification and statistical analysis**. Prism 6.0 (GraphPad Software) and SigmaPlot 14.0 (Systat Software) was used for all statistical analysis. Shapiro–Wilk normality tests were used to determine the appropriate statistical tests to use for analysis, and outliers for taste tests were identified using a robust regression and outlier removal method. Calcium-imaging data were exported to Excel for quantification. Error bars represent means ± s.e.m. A $P$ value of <0.05 was considered statistically significant. For behavioral studies, all $n$ values represent individual mice. For calcium imaging, some $n$ values reflect cell numbers where specified.

**Reporting summary**. Further information on research design is available in the Nature Research Reporting Summary linked to this article.

## Data availability

Other data are available from the corresponding author on reasonable request. Source data are provided with this paper.

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

## Acknowledgements

We thank B. Roth, K. Deisseroth, and L. Zweifel for AAV plasmid constructs, G. Stuber for providing the GCaMP virus, M. Chiang for assistance with animal husbandry, and S. Tsang and K. Kafer for help in generating the *Satb2<sup>Cre</sup>*, *Calca<sup>Cre</sup>*, and *Calca<sup>tdt</sup>* mouse lines. We also thank A. Bowen for sharing data and C. Campos for help with setting up the calcium-imaging experiments. B.C.J. was supported by a National Science Foundation Graduate Research Fellowship (DGE-1256082). J.Y.C. was supported by a National Institute on Drug Abuse fellowship (T32-DA07278). R.D.P. is supported by a National Institutes of Health grant (R01-DA24908). Inscopix provided the calcium-imaging equipment and supplies via the DECODE grant program.

## Author contributions

B.C.J. and J.Y.C. conceived the study and designed experiments. B.C.J., J.Y.C., and H.O.K. performed experiments and data analysis. R.D.P. generated the targeting constructs for *Satb2<sup>Cre</sup>*, *Calca<sup>Cre</sup>*, and *Calca<sup>tdT</sup>* mouse lines. B.C.J. and J.Y.C. wrote the manuscript with input from all authors.

## Competing interests

The authors declare no competing interests.
