## [Peer Review File · Nature Communications]

Reviewers' comments:

Reviewer #1 (Remarks to the Author):

In this manuscript, the authors studied roles of *stab2*-expressing PBN neurons in taste perception. *Stab2* neurons respond to all 5 basic taste stimuli. Activation of *stab2* neurons enhances whereas inhibition of *stab2* neurons attenuate animal's behavioral responses to taste stimuli. The experiments are well executed and data are solid. Nevertheless, I have some suggestions for the authors to consider, in order to strengthen their conclusions.

1. The inactivation experiments in Fig.3 were done by using Tetaus toxin, which effected all downstream targets of the *stab2* neurons. It is interesting to see the impact of selective silencing the thalamic terminals of the *stab2* neurons on taste response.
2. Similarly, what is the behavioral effect of optogenetic activation of the thalamic terminals of the *stab2* neurons on taste response?
3. What is the behavioral response to high concentration of salt after ablating both *stab2* and *Calca* expressing neurons?
4. Optogenetic activation of *stab2* neurons increases sweet consumption but not change Quinine licks is interesting (Fig.4e). What if the author further water deprive the mice to increase the baseline licks of quinine, then test the behavioral effect of optogenetic activation? The two data points from the mice with high baseline quinine licks do decreases their licking when stimulated with light in Fig.4e.

Reviewer #2 (Remarks to the Author):

This manuscript from the Palmiter lab examines the taste coding properties of *Satb2*-expressing neurons in the parabrachial nucleus (PBN). They show that *Satb2* neurons respond to a variety of tastes, usually with broad tuning to tastants. They go on to show that inhibition of these neurons reduces consumption of palatable tastes and increases consumption of unpalatable tastants. Surprisingly, activation of *Satb2* neurons increases taste palatability. Silencing the output of both PBN CGRP neurons and *Satb2* neurons eliminates aversive responses to unpalatable solutions. The experiments are interesting, but the results of imaging and inactivation experiments are not very consistent. This leads to a confusing combination of results that needs to be better addressed by the authors.

1. The heterogenous taste response of these neurons is not well enough explained. For example, it seems that bitter response neurons are relatively less frequent and sweet response are more common than other taste modalities (guessing about this from Fig2 C because there is no color legend for fig 2D). However, the bitter-responsive neurons are mostly overlapping with the sweet-responsive neurons. There is one strongly responding bitter-sensitive neuron in the report.
2. Related to this, the entire imaging study is based on just 54 neurons. The report would be stronger if there were at least 2-3 times this number of cells. It may help resolve what is going on with the bitter coding. As it stands, the in vivo imaging does not support claims about *Satb2* neurons encoding bitter taste.
3. Despite the limited selective responding to bitter taste, there are striking effects of inactivating *Satb2* neurons on avoidance of bitter flavors. More needs to be done to explain this glaring inconsistency.
4. Additional information about the calcium imaging is required. There should be videos supplied showing the dynamic responses before and during consumption so that the imaging quality can be evaluated.
5. Although *Satb2* neurons have many projection targets, the manuscript tends to imply their major contribution through vPMpc to insular cortex. It would be more convincing to show a major contribution for *Satb2* neurons information in vPMpc vs other projection sites. Only showing projection

fibers or CFOS in vPMpc is relatively weak evidence.

6. Claiming lack of thermosensation does not appear to be correct (line 132). 4C water appears to induce lower response amplitudes (supplemental 4d).

7. The signal-to-noise metric appears to be calculated as noise-to-signal. This confusing inconsistency should be dealt with.

8. Z scores are unusually large. Methods say "Z scores were calculated using the formula $(x - \mu_{\text{baseline}}) / \sigma_{\text{baseline}}$, where μ_{baseline} is the average fluorescence and σ_{baseline} is the standard deviation during the 30 s before first lick or stimulus" (line 426-427). Z scores should be calculated as $(x - \text{mean}(\text{whole trace})) / \text{stdev}(\text{whole trace})$.

Reviewer #3 (Remarks to the Author):

Jarvie and colleagues report interesting findings regarding the taste responsiveness and functional significance of SATb2-expressing neurons in the mouse parabrachial nucleus (PBN). SATb2 is a transcription factor expressed selectively in the "waist region" of PBN, a zone long associated with taste responsivity. These findings come close on the heels of a report by Fu et al. (Cell Reports, May 2019) who focused on this same population of cells. Although the two reports agree on some basics (SAT2b neurons are taste neurons), the two studies report many divergent findings and interpretations. Therefore, the present data remain novel. The data are important because, until now, almost no information has been available regarding the functions of phenotypically identified neurons in PBN. Data such as these are essential for untangling the function of different subsets of PBN neurons. In addition to the new data on the SATb2 neurons, the current experiments succeed in concurrently manipulating both the SATb2 and CGRP populations, yielding the intriguing and novel observation that both these cell populations contribute to behavioral rejection of bitter taste.

Overall, the studies were done well and represent a notable technical achievement. However, a number of technical and substantive critical issues require clarification and further analysis to optimize interpretability. One general issue is that it would be useful to expand documentation of the size and location of the various injection sites (synaptophysin, Ca++ imaging, TetTox, and ChR2) by including multiple levels of PBN. Because the mouse PBN is very small and the subnuclear organization changes rapidly over a couple hundred microns, such information could become critical in resolving functions of different subsets of neurons. In addition, the relationship of the present data to other information available on the PBN as well as the other SATb2 study requires more thorough discussion. Detailed comments on these and other issues follow.

Validation of the mouse line

The present study used gene targeting to generate a mouse that expresses cre in SATb2 neurons. The mouse was validated by injecting a cre-dependent YFP virus in PBN and observing that ~90% of YFP+ neurons were immunoreactive for SATb2. However, the manuscript does not specify whether the majority of immunoreactive neurons also express YFP (at least at the center of the injection site). This information is important for evaluating how completely the SATb2 population is targeted.

Projections from SATb2 neurons to the forebrain are mostly similar to PBN projections revealed using conventional tracers or non-cre dependent viral labeling and suggest that, for the most part, they are not unique. However, the projection to the basomedial nucleus is an exception. The magnitude of this projection is surprising based on the earlier data, which show a very sparse projection to this region (e.g., Tokita et al., 2010 and the Allen Brain Atlas in mouse and many rat studies). Even if the current viral labeling is more sensitive, it seems surprising that the projection to the basomedial is as robust as to the central nucleus of the amygdala. This finding deserves comment and discussion.

Gustatory response properties of Satb2 neurons

The authors use Ca⁺⁺ imaging with a cre-dependent GCaMP6s virus to define the sensory response properties of Satb2 neurons. Notably, in contrast to Fu et al. who claim that SATb2 neurons respond only to sweet stimuli, the authors conclude that PBN SATb2 neurons respond to all taste qualities and to water (but not tactile, thermal or olfactory stimuli) and that individual neurons are usually broadly tuned. These divergent findings are notable. However, several aspects of this part of the study make interpretation difficult.

1. I noticed that the coordinates used to target the PBN by Jarvie et al, are about 0.6mm rostral to those used by Fu, a significant distance in the small mouse PBN. Can the authors comment on this difference? Is it possible that this could provide some explanation for the different results? Can the authors provide a figure that would better define the precise level of the PBN where Ca⁺⁺ imaging took place?

2. There is no mention of response reliability over multiple trials. The manuscript indicates that imaging was performed over 3 "sessions" separated by at least an hour, but it is not entirely clear whether individual cells could be identified in these different sessions. This information needs to be included.

3. Although the authors monitored licking while the mice were sampling tastants, no information is available regarding whether licking was comparable for the different fluids. Since the stimuli represent very different hedonic valences, it would be surprising if this were the case (although this is possible if the mice are motivated sufficiently due to deprivation state). If different stimuli elicited different amounts of licking, this may have affected response magnitude. In fact, the authors themselves offer this as an explanation for the lack of a positive response-concentration function for quinine in Supplemental Figure 4. Although differences in licking patterns are clearly not the only determinant of response size (obvious in the simultaneous records in Figure 2b), licking patterns should be illustrated and average lick rates or total licks calculated for each stimulus. Moreover, the authors should perform a supplemental analysis that normalizes taste responses by the amount of licking. In addition, another supplemental analysis that analyzes responses to taste stimuli versus water should be included. Although some evidence has recently emerged suggesting that water may be independent taste, this is still a controversial claim.

4. The tastants chosen represent different qualities but intensities are not well-matched. Most importantly, 1mM saccharin was the sweet stimulus (perhaps this is a typo?). This is a very weak, essentially peri-threshold concentration, as shown by the author's own behavioral data in the brief-access licking test (Fig. 3d) and others (e.g., Treasukosal et al., AJP, 2009). The lack of efficacy of this stimulus is also apparent in neurophysiological studies (e.g., CT: Ohkuri T et al., AJP, 2009). Therefore, it is difficult to evaluate the magnitude or specificity of responses to sweet stimuli. The concentrations of quinine, salt and citric acid are more effective and better matched to each other. However, 10mM MSG is used as the umami stimulus. 10mM MSG by itself constitutes a very weak sodium stimulus. An effective umami stimulus requires a higher concentration of MSG mixed with a ribonucleotide such as IMP. These limitations need to be acknowledged.

5. The supplemental data include sucrose responses at both high and low concentrations and this is helpful. However, in these experiments, only a subset of qualities were tested and the data were not as thoroughly analyzed (for example, averages aren't shown and breadth of tuning is not calculated). This analysis should be expanded. In addition, it is not clear whether these neurons represent a subset of those shown in Fig. 2 (i.e., 1-44 of the 1-54?) or are a different set of neurons recorded on a different day, or perhaps from a different mouse.

6. The manuscript claims no response to thermal stimuli (Supplemental Figure 4D). This result flies in the face of a great deal of previous data showing thermal sensitivity of gustatory afferents (e.g., Ogawa et al., 1968) and central neurons (see Lemon, AJP, 2017 for a review), including the PBN. It is possible that SAT2b neurons are an exception or that the awake, behaving state masks such

responses, but this seems unlikely. In addition, this conclusion is at odds with the data shown in the figure. While there is not a systematic average function with the three temperatures employed, a number of individual neurons show an optimal response at a particular temperature. Given what we know about the relationships between gustatory and thermal sensitivity, heterogeneity is to be expected. It also may be premature to conclude that these neurons do not respond to mechanosensory stimuli given previous reports of taste/mechanical co-sensitivity in PBN (e.g., Ogawa, Hayama, and Ito, '87), as well as other central and peripheral neurons. It may simply be the case that licking an empty spout (how many licks did the mice make; how hard did they lick?), does not constitute an adequate tactile stimulus. Again, these considerations deserve discussion.

Behavioral Functions of SATb2 neurons.

The authors go on to show that inactivation of SATb2 neurons by injecting a cre-dependent virus expressing Tet-Tox attenuates preference/aversion behaviors, as measured in the brief-access lick test and long-term intake tests. Significantly, there were effects across multiple qualities. The data are interesting and convincing, and at odds with the Fu paper which only show effects on the sweet quality. Figure 3a shows the center of an injection site. It would be useful to expand this to multiple panels and to include darkfield images so that the exact location and amount of PBN involved can be better specified; these details could ultimately be significant for resolving discrepancies between studies. It is notable that the authors mention that a possible reason for the lack of effect on citric acid is that the mice seem to be able to smell this stimulus. If possible, for the brief access test, the authors should include a standard analysis of the latency to the first lick for all the stimuli to determine the degree to which smell might have influenced the results (e.g., maybe effects would have been even larger if smell wasn't a factor).

Results from the optogenetic experiments are messier but interesting. For the unconditioned effects shown in Fig. 4 c-e, the figure caption claims that optogenetic stimulation had no effect on water licking, enhanced saccharin licking, but did not further suppress quinine licking. While the conclusion regarding quinine seems consistent with the data in panel e, the individual record in panel c seems to show an effect. Moreover, the statistics presented in the caption claim significant interactions for both the saccharin and quinine data. Am I missing something here? Also, am I correct that the concentrations used are 1mM saccharin and 1mM quinine? If this is the case, perhaps the lack of an effect on quinine is due to floor effect. The flavor conditioning experiments shown in panels f-I, demonstrating effects on both sweet and bitter compounds support such a notion.

The data in Figure 5 showing that deactivation of both Satb2 and CGRP neurons in the external region, using a cre-dependent Tet-Tox virus in a newly generated Satb2-Calca mouse nearly obliterate aversion to quinine and citric acid are compelling. I'm a little puzzled that sweet stimuli were not tested in this paradigm, but presume that such an experiment will be done in the future.

Minor Points & Errors (page numbers are from the "Word" file)

p. 13 Should "indexlens" have a space?

p. 14 "AAV-DJ..." has a "?" in the notation

p. 15 Signal to noise ratio (p. 15) as defined by Spector and Travers should be "noise-to-signal" ratio. Should also change or strike the last sentence in the paragraph since the noise-to-signal ratio only considers responses to 2 stimuli.

p. 15-16 Did the lickometer circuit pass current through the mouse's tongue. If so, how much? The possibility of an "electric taste" sensation may need to be considered (e.g., Smith and Bealer, 1975)

p. 17 "training" should be "trained"

Presumably taste stimuli were presented at room temperature. Please specify.

Both male and female mice were used. I assume that no difference was noted between the sexes, but did not see this mentioned in the text.

Reviewer #1 (Remarks to the Author): **New data are highlighted in yellow**

In this manuscript, the authors studied roles of stab2-expressing PBN neurons in taste perception. Stab2 neurons respond to all 5 basic taste stimuli. Activation of stab2 neurons enhances whereas inhibition of stab2 neurons attenuate animal's behavioral responses to taste stimuli. The experiments are well executed and data are solid. Nevertheless, I have some suggestions for the authors to consider, in order to strengthen their conclusions.

1. The inactivation experiments in Fig.3 were done by using Tetraus toxin, which effected all downstream targets of the stab2 neurons. It is interesting to see the impact of selective silencing the thalamic terminals of the stab2 neurons on taste response.

We agree with the reviewer that inactivating the downstream targets of Satb2 neurons would be a very interesting set of experiments, but feel that analysis of multiple downstream targets would ultimately be the most informative and is outside the scope of the present paper.

2. Similarly, what is the behavioral effect of optogenetic activation of the thalamic terminals of the stab2 neurons on taste response?

We agree with the reviewer that this is an interesting question. However, another group published a paper while this one was under review that did this experiment with Satb2 neurons (Cell Reports: Fu et al., 2019). They showed that animals increased the amount they would lick for water paired with stimulation of Satb2 neuronal terminal in the VPMpc, but this did not occur when activating the projections to the CeA or BNST. While further and more comprehensive analysis of these projections is certainly needed, we again feel that this is outside the scope of the present paper.

3. What is the behavioral response to high concentration of salt after ablating both stab2 and Calca expressing neurons?

We do not have the short-access data after silencing both Satb2 and Calca neurons. To provide a more complete look into the role of Calca neurons in taste, we did add two-bottle preference tests for animals with inactivated Calca neurons. We found that Calca-TetTox mice had a higher preference for both quinine and 0.3M NaCl. These data are **in Fig. 5b**.

4. Optogenetic activation of stab2 neurons increases sweet consumption but not change Quinine licks is interesting (Fig.4e). What if the author further water deprive the mice to increase the baseline licks of quinine, then test the behavioral effect of optogenetic activation? The two data points from the mice with high baseline quinine licks do decreases their licking when stimulated with light in Fig.4e.

We agree that adjusting the parameters of these experiments could reveal an effect on licking for quinine. Unfortunately, mice had already been water deprived overnight and we were not able to deprive them further. We did run the same experiment with a lower concentration of quinine (0.1 mM) so that baseline licks were higher, but there was still no effect. These data were added to **Supplementary Fig. 5b**.

Reviewer #2 (Remarks to the Author):

This manuscript from the Palmiter lab examines the taste coding properties of Satb2-expressing neurons in the parabrachial nucleus (PBN). They show that Satb2 neurons respond to a variety of tastes, usually with broad tuning to tastants. They go on to show that inhibition of these neurons reduces consumption of palatable tastes and increases consumption of unpalatable tastants. Surprisingly, activation of Satb2

neurons increases taste palatability. Silencing the output of both PBN CGRP neurons and *Satb2* neurons eliminates aversive responses to unpalatable solutions. The experiments are interesting, but the results of imaging and inactivation experiments are not very consistent. This leads to a confusing combination of results that needs to be better addressed by the authors.

1. The heterogeneous taste response of these neurons is not well enough explained. For example, it seems that bitter response neurons are relatively less frequent and sweet response are more common than other taste modalities (guessing about this from Fig 2 C because there is no color legend for fig 2D). However, the bitter-responsive neurons are mostly overlapping with the sweet-responsive neurons. There is one strongly responding bitter-sensitive neuron in the report.

We agree that we could more clearly describe the neuronal responses to taste. The most obvious, strongly responding bitter-sensitive neuron had the strongest response out of any neurons for any taste tested, which makes it appear to be the only responding neuron in the heat-map. There were 41/54 bitter-responsive neurons detected in our recording experiments, where the response during licking (normalized to the number of licks) was greater than 3 SD from baseline. Of these, 8 were determined to be bitter-best, where the response to bitter was the greatest compared to other tastes tested. We included a panel that shows the absolute value of the response to each taste that has been normalized to the largest response for that cell. This new figure (Fig. 2e) more accurately shows the number of neurons that responded to each taste. About half of bitter neurons retained their bitter-best identity even at saturating concentrations of sweet solutions, which recruited a greater number of sweet-best neurons (Supplementary Fig 3b-d). We have included a more complete explanation of the *Satb2* responses in both the results and discussion sections of the paper.

2. Related to this, the entire imaging study is based on just 54 neurons. The report would be stronger if there were at least 2-3 times this number of cells. It may help resolve what is going on with the bitter coding. As it stands, the *in vivo* imaging does not support claims about *Satb2* neurons encoding bitter taste.

We agree that the report would be stronger with more neurons but given the extreme technical difficulty of targeting this specific population of cells, we were unable to target these neurons in additional animals to increase the number of cells. An important point is that we refute the conclusion of the Fu et al. (2019) paper, which claimed the *Satb2* neurons only encoded sweet taste (See Figure 2). The conclusions we could potentially draw from adding more cells is unlikely to change from what we have drawn from the current data set.

3. Despite the limited selective responding to bitter taste, there are striking effects of inactivating *Satb2* neurons on avoidance of bitter flavors. More needs to be done to explain this glaring inconsistency. We agree that this is an inconsistency and have updated our figures and text to address this. Please see response to point #1 for discussion of bitter taste response.

Due to the technical limitations of targeting and imaging this population *in vivo*, all of our recording sites tended to be in the same area, which was the waist and medial PBN around -5.5 and -5.6 (now added to Supplementary Fig 3a). Given the robust behavioral effects of inactivating *Satb2* neurons, it is possible that there is topographical organization of *Satb2* neurons, with the more bitter-responsive neurons more rostral or lateral to where we were recording. A similar result has been described by Tokita et al., 2010, showing a greater concentration of bitter-best neurons in the lateral PBN. We have now added this to our discussion.

4. Additional information about the calcium imaging is required. There should be videos supplied showing the dynamic responses before and during consumption so that the imaging quality can be evaluated.

An example video of calcium imaging has been added (Supplementary Video 1).

5. Although Satb2 neurons have many projection targets, the manuscript tends to imply their major contribution through vPMpc to insular cortex. It would be more convincing to show a major contribution for Satb2 neurons information in vPMpc vs other projection sites. Only showing projection fibers or CFOS in vPMpc is relatively weak evidence.

It was not our intent to imply that the major contribution of Satb2 neurons was through the VPMpc, and we have downplayed these parts of the manuscript to avoid making this impression. However, the Fu et al. (2019) paper that came out when we submitted this paper found that projections to the VPMpc were the only ones of those tested that had a behavioral effect. We have updated our discussion to include their finding.

6. Claiming lack of thermosensation does not appear to be correct (line 132). 4C water appears to induce lower response amplitudes (supplemental 4d).

There was a trend at 4C in our original analysis. However, we updated how we analyzed all of our calcium imaging data so that we only looked at the net average response of neurons during the time that animals were licking rather than the 10-s following the first lick. We then normalized the response to the number of licks. When we did this, the previous trend seen at 4C was no longer apparent (Supplementary Fig 3h).

7. The signal-to-noise metric appears to be calculated as noise-to-signal. This confusing inconsistency should be dealt with.

Thank you for catching this. We calculated noise-to-signal and have updated the inconsistencies throughout the manuscript to accurately reflect this.

8. Z scores are unusually large. Methods say “Z scores were calculated using the formula $(x - \mu_{\text{baseline}})/\sigma_{\text{baseline}}$, where μ_{baseline} is the average fluorescence and σ_{baseline} is the standard deviation during the 30 s before first lick or stimulus” (line 426-427). Z scores should be calculated as $(x - \text{mean}(\text{whole trace})/\text{stdev}(\text{whole trace}))$.

We have reanalyzed all the data and present them as $dF/F = (F - F_0)/F_0$.

Reviewer #3 (Remarks to the Author):

Jarvie and colleagues report interesting findings regarding the taste responsiveness and functional significance of SATb2-expressing neurons in the mouse parabrachial nucleus (PBN). SATb2 is a transcription factor expressed selectively in the “waist region” of PBN, a zone long associated with taste responsivity. These findings come close on the heels of a report by Fu et al. (Cell Reports, May 2019) who focused on this same population of cells. Although the two reports agree on some basics (SAT2b neurons are taste neurons), the two studies report many divergent findings and interpretations. Therefore, the present data remain novel. The data are important because, until now, almost no information has been available regarding the functions of phenotypically identified neurons in PBN. Data such as these are essential for untangling the function of different subsets of PBN neurons. In addition

to the new data on the SATb2 neurons, the current experiments succeed in concurrently manipulating both the SATb2 and CGRP populations, yielding the intriguing and novel observation that both these cell populations contribute to behavioral rejection of bitter taste.

Overall, the studies were done well and represent a notable technical achievement. However, a number of technical and substantive critical issues require clarification and further analysis to optimize interpretability. One general issue is that it would be useful to expand documentation of the size and location of the various injection sites (synaptophysin, Ca⁺⁺ imaging, TetTox, and ChR2) by including multiple levels of PBN. Because the mouse PBN is very small and the subnuclear organization changes rapidly over a couple hundred microns, such information could become critical in resolving functions of different subsets of neurons. In addition, the relationship of the present data to other information available on the PBN as well as the other SATb2 study requires more thorough discussion. Detailed comments on these and other issues follow.

We have included a more complete discussion of our data relative to other information available on the PBN as well as the other Satb2 study.

Validation of the mouse line

The present study used gene targeting to generate a mouse that expresses cre in SATb2 neurons. The mouse was validated by injecting a cre-dependent YFP virus in PBN and observing that ~90% of YFP+ neurons were immunoreactive for SATb2. However, the manuscript does not specify whether the majority of immunoreactive neurons also express YFP (at least at the center of the injection site). This information is important for evaluating how completely the SATb2 population is targeted.

We agree that more detail about the percentage and location of Satb2 neurons targeted by our viral injections is important for interpreting the data. We have added the number of Sab2 neurons labeled with YFP from the validation experiments, as well as a more detailed graph and pictures showing viral transduction throughout the extent of the PBN (Supplementary Figure 1b-c). The majority of Satb2 neurons were labeled with virus (~74% total), particularly in the middle to caudal portion of the PBN where the bulk of the Satb2 neurons reside. The most rostral Satb2 neurons were not as completely covered, although these are a much smaller portion of the population.

We did not quantify targeting efficiency in all the experiments using viral injections. However, the images shown are representative of what we considered to be a “hit” and required for inclusion of an animal in a behavioral data set. A further example of this can be seen in Supplementary Fig 5a, where we show multiple images with viral expression of ChR2:YFP and Fos from our optogenetic experiments.

Projections from SATb2 neurons to the forebrain are mostly similar to PBN projections revealed using conventional tracers or non-cre dependent viral labeling and suggest that, for the most part, they are not unique. However, the projection to the basomedial nucleus is an exception. The magnitude of this projection is surprising based on the earlier data, which show a very sparse projection to this region (e.g., Tokita et al., 2010 and the Allen Brain Atlas in mouse and many rat studies). Even if the current viral labeling is more sensitive, it seems surprising that the projection to the basomedial is as robust as to the central nucleus of the amygdala. This finding deserves comment and discussion.

We thank the reviewer for this insight. While it is true that other groups have identified PBN projections to the BMA, these are usually weaker than those to the CeA. Satb2 neurons do appear to send equally strong projections to both the CeA and BMA, suggesting that the BMA projection is unique to Satb2

neurons compared to the rest of the populations in the PBN. We have updated our discussion to highlight this finding.

Gustatory response properties of Satb2 neurons

The authors use Ca⁺⁺ imaging with a cre-dependent GCaMP6s virus to define the sensory response properties of Satb2 neurons. Notably, in contrast to Fu et al. who claim that SATb2 neurons respond only to sweet stimuli, the authors conclude that PBN SATb2 neurons respond to all taste qualities and to water (but not tactile, thermal or olfactory stimuli) and that individual neurons are usually broadly tuned. These divergent findings are notable. However, several aspects of this part of the study make interpretation difficult.

1. I noticed that the coordinates used to target the PBN by Jarvie et al, are about 0.6mm rostral to those used by Fu, a significant distance in the small mouse PBN. Can the authors comment on this difference? Is it possible that this could provide some explanation for the different results? Can the authors provide a figure that would better define the precise level of the PBN where Ca⁺⁺ imaging took place?

Our targeting tends to be more caudal than the Atlas coordinates, hence the more rostral numerical coordinates. We have added images of lens placement from our calcium imaging experiments to Supplementary Fig. 3a, which are in a very similar but maybe slightly rostral area to the area shown by Fu et al (2019).

2. There is no mention of response reliability over multiple trials. The manuscript indicates that imaging was performed over 3 “sessions” separated by at least an hour, but it is not entirely clear whether individual cells could be identified in these different sessions. This information needs to be included.

Examples of the response of individual neurons to the presentation of the same taste across the 3 separate sessions in the same day is now shown in Fig 2c.

3. Although the authors monitored licking while the mice were sampling tastants, no information is available regarding whether licking was comparable for the different fluids. Since the stimuli represent very different hedonic valences, it would be surprising if this were the case (although this is possible if the mice are motivated sufficiently due to deprivation state). If different stimuli elicited different amounts of licking, this may have affected response magnitude. In fact, the authors themselves offer this as an explanation for the lack of a positive response-concentration function for quinine in Supplemental Figure 4. Although differences in licking patterns are clearly not the only determinant of response size (obvious in the simultaneous records in Figure 2b), licking patterns should be illustrated and average lick rates or total licks calculated for each stimulus. Moreover, the authors should perform a supplemental analysis that normalizes taste responses by the amount of licking. In addition, another supplemental analysis that analyzes responses to taste stimuli versus water should be included. Although some evidence has recently emerged suggesting that water may be independent taste, this is still a controversial claim.

Thank you for these suggestions. We now show calcium responses relative to licks for example cells in Fig 2c. All our data have been updated to reflect the response of neurons during the time the animal was licking, rather than for 10-s following the first lick. This was then normalized to the number of licks. We believe that this is a much better way to control for different sampling sizes and different hedonic values between stimuli.

Our additional analysis of data in Fig 2e shows that some neurons clearly respond best to water. When we normalize these responses to water, the water-best neurons are simply redistributed to different

categories, while relative responses from all other cells remained the same. As this did not add further information to our analysis, we feel that normalizing to sampling time and lick number is a much cleaner way to present the data that does not make assumptions as to whether water is an independent taste.

4. The tastants chosen represent different qualities but intensities are not well-matched. Most importantly, 1mM saccharin was the sweet stimulus (perhaps this is a typo?). This is a very weak, essentially peri-threshold concentration, as shown by the author's own behavioral data in the brief-access licking test (Fig. 3d) and others (e.g., Treesukosal et al., AJP, 2009). The lack of efficacy of this stimulus is also apparent in neurophysiological studies (e.g., CT: Ohkuri T et al., AJP, 2009). Therefore, it is difficult to evaluate the magnitude or specificity of responses to sweet stimuli. The concentrations of quinine, salt and citric acid are more effective and better matched to each other. However, 10mM MSG is used as the umami stimulus. 10mM MSG by itself constitutes a very weak sodium stimulus. An effective umami stimulus requires a higher concentration of MSG mixed with a ribonucleotide such as IMP. These limitations need to be acknowledged.

Thank you for pointing this out, we have included the limitations of MSG as an umami stimulus to our discussion section. A low concentration of saccharin was chosen because the mice were water deprived and we did not want the preference for sweet tastes to interfere with licking for other tastes during testing. Our analysis of sweet taste responses in Supplementary Fig. 3b-d highlights the specificity and magnitude of responses to sucrose at both low and high concentrations.

5. The supplemental data include sucrose responses at both high and low concentrations and this is helpful. However, in these experiments, only a subset of qualities were tested and the data were not as thoroughly analyzed (for example, averages aren't shown and breadth of tuning is not calculated). This analysis should be expanded. In addition, it is not clear whether these neurons represent a subset of those shown in Fig. 2 (i.e., 1-44 of the 1-54?) or are a different set of neurons recorded on a different day, or perhaps from a different mouse.

We have expanded our analysis of the low vs high concentrations of taste solutions. The averages of all taste response neurons are shown in Supplementary Fig. 3b. Additionally, we have plotted each neuron's best response at low and high concentrations, which better illustrates the change in best-response when tastes were presented at high concentrations. We also added an analysis of the magnitude of taste responses by projecting the responses at low concentrations onto the high, which further quantifies the change in response magnitude for each taste and concentration (Supplementary Fig. 3c-d). We found that while many neurons become sweet-best at high concentrations of sweet, some neurons retain their bitter or salt-best categorization.

Experiments were done on a different day from those shown in Fig. 2, so we cannot say with certainty that the 43 neurons we recorded from are a subset or entirely different from the 54 initial neurons. All experiments include data from all 3 mice.

6. The manuscript claims no response to thermal stimuli (Supplemental Figure 4D). This result flies in the face of a great deal of previous data showing thermal sensitivity of gustatory afferents (e.g., Ogawa et al., 1968) and central neurons (see Lemon, AJP, 2017 for a review), including the PBN. It is possible that SAT2b neurons are an exception or that the awake, behaving state masks such responses, but this seems unlikely. In addition, this conclusion is at odds with the data shown in the figure. While there is not a systematic average function with the three temperatures employed, a number of individual neurons show an optimal response at a particular temperature. Given what we know about the relationships between gustatory and thermal sensitivity, heterogeneity is to be expected.

When we updated the data to only include calcium responses during the time that the animals are licking rather than during the 10-s following the first lick and normalized this to the number of licks, the potential difference in response of some neurons at 4C was no longer apparent. We do not have an explanation for why these neurons do not seem to encode temperature, and it is possible that a subset of Satb2 we did not record from do respond to temperature. We have updated our discussion to reflect this.

It also may be premature to conclude that these neurons do not respond to mechanosensory stimuli given previous reports of taste/mechanical co-sensitivity in PBN (e.g., Ogawa, Hayama, and Ito, '87), as well as other central and peripheral neurons. It may simply be the case that licking an empty spout (how many licks did the mice make; how hard did they lick?), does not constitute an adequate tactile stimulus. Again, these considerations deserve discussion.

We agree with the reviewer that we cannot definitively say that Satb2 neurons do not respond to mechanosensory stimuli. For the experiments in this paper, we cannot determine whether mice licked less hard for an empty bottle, but their licking was sufficient to be detected by our lickometer and we have added the average trial lick numbers to Supplementary Fig 3i. It is possible that Satb2 neurons could respond to an alternative tactile stimulus, or that there are some Satb2 neurons that do respond to mechanical stimuli that were not captured in our recordings. We have updated the discussion to reflect this.

Behavioral Functions of SATb2 neurons.

The authors go on to show that inactivation of SATb2 neurons by injecting a cre-dependent virus expressing Tet-Tox attenuates preference/aversion behaviors, as measured in the brief-access lick test and long-term intake tests. Significantly, there were effects across multiple qualities. The data are interesting and convincing, and at odds with the Fu paper which only show effects on the sweet quality. Figure 3a shows the center of an injection site. It would be useful to expand this to multiple panels and to include darkfield images so that the exact location and amount of PBN involved can be better specified; these details could ultimately be significant for resolving discrepancies between studies. It is notable that the authors mention that a possible reason for the lack of effect on citric acid is that the mice seem to be able to smell this stimulus. If possible, for the brief access test, the authors should include a standard analysis of the latency to the first lick for all the stimuli to determine the degree to smell might have influenced the results (e.g., maybe effects would have been even larger if smell wasn't a factor).

We have included an example of the extent of viral transduction we saw for our viral targeting and what was used qualitatively to determine whether each animal was a viral "hit" into Supplementary Fig. 1b-c. There is also an example of Fos induction with Chr2 stimulation at different rostral-caudal level of the PBN in Supplementary Fig 5a. Please see response to early point about mouse line validation for more details.

We removed the discussion of odor having a potential role in masking an effect for sour taste. We did not have quantitative data to support this, and animals completed a reasonable number of trials despite the observations of the authors while running the experiments. In addition, inactivating both Satb2 and CGRP neurons simultaneously had a very robust effect on licking for sour, which does imply that both populations have a role in sour taste. As we cannot say for certain whether mice with inactivation of CGRP alone also showed a response to sour odor, and we do not have evidence that CGRP neurons

respond to or are critical for response to odor, we did not feel comfortable making this supposition in our discussion.

Results from the optogenetic experiments are messier but interesting. For the unconditioned effects shown in Fig. 4 c-e, the figure caption claims that optogenetic stimulation had no effect on water licking, enhanced saccharin licking, but did not further suppress quinine licking. While the conclusion regarding quinine seems consistent with the data in panel e, the individual record in panel c seems to show an effect. Moreover, the statistics presented in the caption claim significant interactions for both the saccharin and quinine data. Am I missing something here?

As noted by the reviewer, the licking behavior is rather variable and we averaged responses across multiple trials to control for this. We did look for a better representative trace, but the record in panel c is representative of a single licking trial by the mouse.

There is a significant effect between water and quinine in the brief-access tests, which just indicated that mice licked less for quinine than water. However, stimulation of *Satb2* neurons did not change licking for either quinine or water in these trials. We have updated how these data in Figure 4 to make the results easier to interpret (Fig 4d,e).

Also, am I correct that the concentrations used are 1mM saccharin and 1mM quinine? If this is the case, perhaps the lack of an effect on quinine is due to floor effect. The flavor conditioning experiments shown in panels f-l, demonstrating effects on both sweet and bitter compounds support such a notion.

The concentrations used for the brief-access tests were 1 mM saccharin and 0.3 mM quinine, and we have updated the figure legend to more clearly state this. We also added data for the same experiment only run with 0.1 mM quinine to look for a floor effect, but there was no effect on licking for this lower concentration of quinine during stimulation of *Satb2* neurons (Supplementary Fig 5b).

The data in Figure 5 showing that deactivation of both *Satb2* and CGRP neurons in the external region, using a cre-dependent Tet-Tox virus in a newly generated *Satb2*-*Calca* mouse nearly obliterate aversion to quinine and citric acid are compelling. I'm a little puzzled that sweet stimuli were not tested in this paradigm, but presume that such an experiments will be done in the future.

We have added both two-bottle choice and brief-access data looking at saccharin intake in *Calca* mice. There were no differences in intake or willingness to lick for saccharin in TetTox mice compared to control animals. These experiments have been added to the text and Fig. 5b. As there was no effect of *Calca* inactivation on licks for saccharin, we did not pursue an additive effect in in the *Satb2::Calca*-TetTox mice.

Minor Points & Errors (page numbers are from the "Word" file)

p. 13 Should "indexlens" have a space?

p. 14 "AAV-DJ..." has a "?" in the notation

p. 15 Signal to noise ratio (p. 15) as defined by Spector and Travers should be "noise-to-signal" ratio. Should also change or strike the last sentence in the paragraph since the noise-to-signal ratio only considers responses to 2 stimuli.

These changes have been made to the text.

p. 15-16 Did the lickometer circuit pass current through the mouse's tongue. If so, how much? The possibility of an "electric taste" sensation may need to be considered (e.g., Smith and Bealer, 1975) The lickometer passes less than 60 nA through the tongue of the mouse, and we have added this detail to the methods section. Mice do not appear capable of detecting currents less than 1 μ A (Gannon et al., 1992), and we failed to see any neuronal responses to licking in the absence of taste (empty bottle experiment in Supplementary Fig 4c; the neuronal responses have now been normalized to the number of licks). Collectively this suggests that "electric taste" sensation is not a factor in these experiments.

Gannon KS, Smith JC, Henderson R, Hendrick P. A system for studying the microstructure of ingestive behavior in mice. *Physiol Behav.* 1992;51(3):515-521. doi:10.1016/0031-9384(92)90173-y

p. 17 "training" should be "trained"

Presumably taste stimuli were presented at room temperature. Please specify.

Yes, taste stimuli were presented at room temperature and this had been added to the methods section.

Both male and female mice were used. I assume that no difference was noted between the sexes, but did not see this mentioned in the text.

We did not do a formal comparison between male and female mice, but there were no obvious differences apparent during testing. We have added this to the methods section.

REVIEWER COMMENTS

Reviewer #1 (Remarks to the Author):

The authors have properly addressed my questions. I have no more concerns on this manuscript.

Reviewer #2 (Remarks to the Author):

The authors have addressed my concerns. One typo: Line 97 should be VPMpc

Reviewer #3 (Remarks to the Author):

I thank the authors for their revision of this interesting paper and apologize that this review is late. The authors have addressed all my concerns about the behavioral/activation and inactivation experiments. These data provide convincing and important data that Sat2b neurons contribute to the processing of multiple taste qualities and that they work together with CGRP neurons in underlying responses to aversive tastes.

I also had a number of concerns about the Ca⁺⁺ imaging data. These have been partially addressed but there are still important issues with these data. These are not trivial concerns, because they are key for several controversial issues regarding taste quality coding.

Given the low hit rate for the Ca⁺⁺ imaging studies, I appreciate the difficulty of collecting these data and realize that more experiments are not likely to be forthcoming. However, the authors need to provide:

- (1) A fuller description of the consistency of the reported responses in Figure 2. The single example that was provided in the revision looks stable, but is not sufficient to allow assessment of the population. I would suggest that Figure 2e be expanded to present mean (summed or average) responses to each stimulus for each cell (e.g., as in Tokita and Boughter, *J Neurofizz*, 2012) with symbols representing individual trials superimposed on the mean responses. This would also make it easier to evaluate response profiles for the neurons.
- (2) It is still very puzzling to me that neither thermal nor mechansensory stimuli activated these cells, given the many prior reports that such stimuli are efficacious for PBN taste neurons. Certainly it is possible that the Satb2 neurons are an exception but it would be reassuring to show that during the sessions where dry bottle licking and thermal stimuli were tested, that these same cells still responded to taste. Perhaps a control taste stimulation is available for each of these sessions?
- (3) I understand that the authors chose to use a weak sweet stimulus (1mM saccharin) for a reason. However, some explicit mention of this (with behavioral and neural citations from the literature) is warranted since this is the representative sweet stimulus included in the main Ca⁺⁺ imaging results reported in (Fig 2) in the body of the paper. How do the authors explain a robust response to a stimulus that is essentially perithreshold? I find it interesting that the cells in supplemental Figure 3 look so much more narrowly tuned when a more robust concentration of sucrose is employed.

Minor

The photomicrographs showing viral expression and lens placement in Figure 3 are a welcome addition. However, without brightening the image, it is difficult to discern the location. Is a counterstain available on another channel? If not, perhaps a darkfield image could be paired with each fluorescent image, or at least the BC could be dotted in.

Reviewer #1 (Remarks to the Author):

The authors have properly addressed my questions. I have no more concerns on this manuscript.

Reviewer #2 (Remarks to the Author):

The authors have addressed my concerns. One typo: Line 97 should be VPMpc
We thank the reviewer for catching this typo. It has been corrected.

Reviewer #3 (Remarks to the Author): **New data are highlighted in yellow**

I thank the authors for their revision of this interesting paper and apologize that this review is late. The authors have addressed all my concerns about the behavioral/activation and inactivation experiments. These data provide convincing and important data that Sat2b neurons contribute to the processing of multiple taste qualities and that they work together with CGRP neurons in underlying responses to aversive tastes.

I also had a number of concerns about the Ca⁺⁺ imaging data. These have been partially addressed but there are still important issues with these data. These are not trivial concerns, because they are key for several controversial issues regarding taste quality coding.

Given the low hit rate for the Ca⁺⁺ imaging studies, I appreciate the difficulty of collecting these data and realize that more experiments are not likely to be forthcoming. However, the authors need to provide:

(1) A fuller description of the consistency of the reported responses in Figure 2. The single example that was provided in the revision looks stable but is not sufficient to allow assessment of the population. I would suggest that Figure 2e be expanded to present mean (summed or average) responses to each stimulus for each cell (e.g., as in Tokita and Boughter, J Neurofizz, 2012) with symbols representing individual trials superimposed on the mean responses. This would also make it easier to evaluate response profiles for the neurons.

These data have been added as **Supplementary Figure 3b; they show average responses of individual Satb2 neurons to different tastants across three trials with symbols representing responses from each trial.**

(2) It is still very puzzling to me that neither thermal nor mechansensory stimuli activated these cells, given the many prior reports that such stimuli are efficacious for PBN taste neurons. Certainly it is possible that the Satb2 neurons are an exception but it would be reassuring to show that during the sessions where dry bottle licking and thermal stimuli were tested, that these same cells still responded to taste. Perhaps a control taste stimulation is available for each of these sessions?

Indeed, these experiments were done on the same day as the concentration study, with several hours in between recording sessions. We have added the Satb2 neuron responses to 500 mM sucrose to show that these cells still responded to taste. The new heat map is in **Supplementary Figure 4g.**

(3) I understand that the authors chose to use a weak sweet stimulus (1mM saccharin) for a reason. However, some explicit mention of this (with behavioral and neural citations from the literature) is

warranted since this is the representative sweet stimulus included in the main Ca⁺⁺ imaging results reported in (Fig 2) in the body of the paper. How do the authors explain a robust response to a stimulus that is essentially perithreshold? I find it interesting that the cells in supplemental Figure 3 look so much more narrowly tuned when a more robust concentration of sucrose is employed.

We have included a sentence in the discussion identifying that 1 mM saccharin is a weak/perithreshold sweet stimulus with citations from the literature.

We do not have a clear explanation for why we saw a robust response for this concentration of saccharin. Mice were still able to clearly differentiate between water and 1 mM saccharin in two-bottle choice tests (Fig 3f), and the time-locked nature of the responses of Satb2 neurons to licking for 1 mM saccharin indicates that Satb2 neurons receive information about this weak concentration of saccharin despite not showing a clear preference in the brief access tests.

Minor

The photomicrographs showing viral expression and lens placement in Figure 3 are a welcome addition. However, without brightening the image, it is difficult to discern the location. Is a counterstain available on another channel? If not, perhaps a darkfield image could be paired with each fluorescent image, or at least the BC could be dotted in.

We have added outlines of relevant landmarks and included a schematic of approximate PBN location next to the images. Please see the updates in Supplementary Figure 3a.

REVIEWERS' COMMENTS

Reviewer #3 (Remarks to the Author):

The reviewers have now answered all my remaining concerns about the Ca⁺⁺ imaging data, having provided important additional information important for interpretation. I enjoyed reviewing this very interesting paper.